# Minimax Estimation of Neural Net Distance

**Kaiyi Ji**
Department of ECE
The Ohio State University
Columbus, OH 43210
ji.367@osu.edu

**Yingbin Liang**
Department of ECE
The Ohio State University
Columbus, OH 43210
liang.889@osu.edu

## Abstract

An important class of distance metrics proposed for training generative adversarial networks (GANs) is the integral probability metric (IPM), in which the neural net distance captures the practical GAN training via two neural networks. This paper investigates the minimax estimation problem of the neural net distance based on samples drawn from the distributions. We develop the first known minimax lower bound on the estimation error of the neural net distance, and an upper bound tighter than an existing bound on the estimator error for the empirical neural net distance. Our lower and upper bounds match not only in the order of the sample size but also in terms of the norm of the parameter matrices of neural networks, which justifies the empirical neural net distance as a good approximation of the true neural net distance for training GANs in practice.

## 1 Introduction

Generative adversarial networks (GANs), first introduced by [9], have become an important technique for learning generative models from complicated real-life data. Training GANs is performed via a minmax optimization with the maximum and minimum respectively taken over a class of discriminators and a class of generators, where both discriminator and generators are modeled by neural networks. Given that the discriminator class is sufficiently large, [9] interpreted the GAN training as finding a generator such that the generated distribution $\nu$ is as close as possible to the target true distribution $\mu$, measured by the Jensen-Shannon distance $d_{JS}(\mu, \nu)$, as shown below:

$$\min_{\nu \in \mathcal{D}_{\mathcal{G}}} d_{JS}(\mu, \nu). \tag{1}$$

Inspired by such an idea, a large body of GAN models were then proposed based on various distance metrics between a pair of distributions, in order to improve the training stability and performance, e.g., [2, 3, 13, 15]. Among them, the integral probability metric (IPM) [19] arises as an important class of distance metrics for training GANs, which takes the following form

$$d_{\mathcal{F}}(\mu, \nu) = \sup_{f \in \mathcal{F}} \left| \mathbb{E}_{x \sim \mu} f(x) - \mathbb{E}_{x \sim \nu} f(x) \right|. \tag{2}$$

In particular, different choices of the function class $\mathcal{F}$ in (2) result in different distance metrics. For example, if $\mathcal{F}$ represents a set of all 1-Lipschitz functions, then $d_{\mathcal{F}}(\mu, \nu)$ corresponds to the Wasserstein-1 distance, which is used in Wasserstein-GAN (WGAN) [2]. If $\mathcal{F}$ represents a unit ball in a reproducing kernel Hilbert space (RKHS), then $d_{\mathcal{F}}(\mu, \nu)$ corresponds to the maximum mean discrepancy (MMD) distance, which is used in MMD-GAN [7, 13].

Practical GAN training naturally motivates to take $\mathcal{F}$ in (2) as a set $\mathcal{F}_{nn}$ of neural networks, which results in the so-called *neural net distance* $d_{\mathcal{F}_{nn}}(\mu, \nu)$ introduced and studied in [3, 28]. For computational feasibility, in practice $d_{\mathcal{F}_{nn}}(\hat{\mu}_n, \hat{\nu}_m)$ is typically adopted as an approximation (i.e., an estimator) of the true neural net distance $d_{\mathcal{F}_{nn}}(\mu, \nu)$ for the practical GAN training, where $\hat{\mu}_n$

and $\hat{\nu}_m$ are the empirical distributions corresponding to $\mu$ and $\nu$, respectively, based on $n$ samples drawn from $\mu$ and $m$ samples drawn from $\nu$. Thus, one important question one can ask here is how well $d_{\mathcal{F}_{nn}}(\hat{\mu}_n, \hat{\nu}_m)$ approximates $d_{\mathcal{F}_{nn}}(\mu, \nu)$. If they are close, then training GANs to small $d_{\mathcal{F}_{nn}}(\hat{\mu}_n, \hat{\nu}_m)$ also implies small $d_{\mathcal{F}_{nn}}(\mu, \nu)$, i.e., the generated distribution $\nu$ is guaranteed to be close to the true distribution $\mu$.

To answer this question, [3] derived an upper bound on the quantity $|d_{\mathcal{F}_{nn}}(\mu, \nu) - d_{\mathcal{F}_{nn}}(\hat{\mu}_n, \hat{\nu}_m)|$, and showed that $d_{\mathcal{F}_{nn}}(\hat{\mu}_n, \hat{\nu}_m)$ converges to $d_{\mathcal{F}_{nn}}(\mu, \nu)$ at a rate of $\mathcal{O}(n^{-1/2} + m^{-1/2})$. However, the following two important questions are still left open: (a) Whether the rate $O(n^{-1/2} + m^{-1/2})$ of convergence is optimal? We certainly want to be assured that the empirical objective $d_{\mathcal{F}_{nn}}(\hat{\mu}_n, \hat{\nu}_m)$ used in practice does not fall short at the first place. (b) The dependence of the upper bound on neural networks in [3] is characterized by the total number of parameters of neural networks, which can be quite loose by considering recent work, e.g., [20, 27]. Thus, the goal of this paper is to address the above issue (a) by developing a lower bound on the *minimax* estimation error of $d_{\mathcal{F}_{nn}}(\mu, \nu)$ (see Section 2.2 for the precise formulation) and to address issue (b) by developing a tighter upper bound than [3].

In fact, the above problem can be viewed as a distance estimation problem, i.e., estimating the neural net distance $d_{\mathcal{F}_{nn}}(\mu, \nu)$ based on samples i.i.d. drawn from $\mu$ and $\nu$, respectively. The empirical distance $d_{\mathcal{F}_{nn}}(\hat{\mu}_n, \hat{\nu}_m)$ serves as its *plug-in* estimator (i.e., substituting the true distributions by their empirical versions). We are interested in exploring the optimality of the convergence of such a plug-in estimator not only in terms of the size of samples but also the parameters of neural networks. We further note that the neural net distance can be used in a variety of other applications such as the support measure machine [18] and the anomaly detection [29], and hence the performance guarantee we establish here can be of interest in those domains.

## 1.1 Our Contribution

In this paper, we investigate the minimax estimation of the neural net distance $d_{\mathcal{F}_{nn}}(\mu, \nu)$, where the major challenge in analysis lies in dealing with complicated neural network functions. This paper establishes a tighter upper bound on the convergence rate of the empirical estimator than the existing one in [3], and develop a lower bound that matches our upper bound not only in the order of the sample size but also in terms of the norm of the parameter matrices of neural networks. Our specific contributions are summarized as follows:

- In Section 3.1, we provide the first known lower bound on the minimax estimation error of $d_{\mathcal{F}_{nn}}(\mu, \nu)$ based on finite samples, which takes the form as $c_l \max\left(n^{-1/2}, m^{-1/2}\right)$ where the constant $c_l$ depends only on the parameters of neural networks. Such a lower bound further specializes to $b_l \prod_{i=1}^d M(i) \max\left(n^{-1/2}, m^{-1/2}\right)$ for ReLU networks, where $b_l$ is a constant, $d$ is the depth of neural networks and $M(i)$ can be either the Frobenius norm or $\|\cdot\|_{1,\infty}$ norm constraint of the parameter matrix $\mathbf{W}_i$ in layer $i$. Our proof exploits the Le Cam's method with the technical development of a lower bound on the difference between two neural networks.

- In Section 3.2, we develop an upper bound on the estimation error of $d_{\mathcal{F}_{nn}}(\mu, \nu)$ by $d_{\mathcal{F}_{nn}}(\hat{\mu}_n, \hat{\nu}_m)$, which takes the form as $c_u(n^{-1/2} + m^{-1/2})$, where the constant $c_u$ depends only on the parameters of neural networks. Such an upper bound further specializes to $b_u \sqrt{d+h} \prod_{i=1}^d M(i)(n^{-1/2} + m^{-1/2})$ for ReLU networks, where $b_u$ is a constant, $h$ is the dimension of the support of $\mu$ and $\nu$, and $\sqrt{d+h}$ can be replaced by $\sqrt{d}$ or $\sqrt{d + \log h}$ depending on the distribution class and the norm of the weight matrices. Our proof includes the following two major technical developments presented in Section 3.4.

  - A new concentration inequality: In order to develop an upper bound for the unbounded-support sub-Gaussican class, standard McDiarmid inequality under bounded difference condition is not applicable. We thus first generalize a McDiarmid inequality [11] for unbounded functions of *scalar* sub-Gaussian variables to that of sub-Gaussian *vectors*, which can be of independent interest for other applications. Such a development requires substantial machineries. We then apply such a concentration inequality to upper-bounding the estimation error of the neural net distance in terms of Rademacher complexity.

  - Upper bound on Rademacher complexity: Though existing Rademacher complexity bounds [8, 22] of neural networks can be used for input data with bounded support, direct applications of those bounds to the unbounded sub-Gaussian input data yield order-level loose

bounds. Thus, we develop a tighter bound on the Rademacher complexity that exploits the sub-Gaussianity of the input variables. Such a bound is also tighter than the existing same type by [23]. The details of the comparison are provided after Theorem 7.

- In Section 3.3, comparison of the lower and upper bounds indicates that the empirical neural net distance (i.e., the plug-in estimator) achieves the optimal minimax estimation rate in terms of $n^{-1/2} + m^{-1/2}$. Furthermore, for ReLU networks, the two bounds also match in terms of $\prod_{i=1}^d M(i)$, indicating that both $\prod_{i=1}^d \|\mathbf{W}_i\|_F$ and $\prod_{i=1}^d \|\mathbf{W}_i\|_{1,\infty}$ are key quantities that capture the estimation accuracy. Such a result is consistent with those made in [20] for the generalization error of training deep neural networks. We note that there is still a gap $\sqrt{d}$ between the bounds, which requires future efforts to address.

## 1.2 Related Work

**Estimation of IPMs.** [25] studied the empirical estimation of several IPMs including the Wasserstein distance, MMD and Dudley metric, and established the convergence rate for their empirical estimators. A recent paper [26] established that the empirical estimator of MMD achieves the minimax optimal convergence rate. [3] introduced the neural net distance that also belongs to the IPM class, and established the convergence rate of its empirical estimator. This paper establishes a tighter upper bound for such a distance metric, as well as a lower bound that matches our upper bound in the order of sample sizes and the norm of the parameter matrices.

**Generalization error of GANs.** In this paper, we focus on the minimax estimation error of the neural net distance, and hence the quantity $|d_{\mathcal{F}_{nn}}(\mu, \nu) - d_{\mathcal{F}_{nn}}(\hat{\mu}_n, \hat{\nu}_m)|$ is of our interest, on which our bound is tighter than the earlier study in [3]. Such a quantity relates but is different from the following generalization error recently studied in [14, 28] for training GANs. [28] studied the generalization error $d_{\mathcal{F}}(\mu, \hat{\nu}^*) - \inf_{\nu \in \mathcal{D}_{\mathcal{G}}} d_{\mathcal{F}}(\mu, \nu)$, where $\hat{\nu}^*$ was the minimizer of $d_{\mathcal{F}}(\hat{\mu}_n, \nu)$ and $\mathcal{F}$ was taken as a class $\mathcal{F}_{nn}$ of neural networks. [14] studied the same type of generalization error but took $\mathcal{F}$ as a Sobolev space, and characterized how the smoothness of Sobolev space helps the GAN training.

**Rademacher complexity of neural networks.** Part of our analysis of the minimax estimation error of the neural net distance requires to upper-bound the *average* Rademacher complexity of neural networks. Although various bounds on the Rademacher complexity of neural networks, e.g., [1, 4, 8, 21, 22], can be used for distributions with bounded support, direct application of the best known existing bound for sub-Gaussian variables turns out to be order-level looser than the bound we establish here. [6, 23] studied the average Rademacher complexity of one-hidden layer neural networks over Gaussian variables. Specialization of our bound to the setting of [23] improves its bound, and to the setting of [6] equals its bound.

## 1.3 Notations

We use the bold-faced small and capital letters to denote vectors and matrices, respectively. Given a vector $\mathbf{w} \in \mathbb{R}^h$, $\|\mathbf{w}\|_2 = \left(\sum_{i=1}^h \mathbf{w}_i^2\right)^{1/2}$ denotes the $\ell_2$ norm, and $\|\mathbf{w}\|_1 = \sum_{i=1}^h |\mathbf{w}_i|$ denotes the $\ell_1$ norm, where $\mathbf{w}_i$ denotes the $i^{th}$ coordinate of $\mathbf{w}$. For a matrix $\mathbf{W} = [\mathbf{W}_{ij}]$, we use $\|\mathbf{W}\|_F = \left(\sum_{i,j} \mathbf{W}_{ij}^2\right)^{1/2}$ to denote its Frobenius norm, $\|\mathbf{W}\|_{1,\infty}$ to denote the maximal $\ell_1$ norm of the row vectors of $\mathbf{W}$, and $\|\mathbf{W}\|$ to denote its spectral norm. For a real distribution $\mu$, we denote $\hat{\mu}_n$ as its empirical distribution, which takes $1/n$ probability on each of the $n$ samples i.i.d. drawn from $\mu$.

## 2 Preliminaries and Problem Formulations

In this section, we first introduce the neural net distance and the specifications of the corresponding neural networks. We then introduce the minimax estimation problem that we study in this paper.

### 2.1 Neural Net Distance

The neural net distance between two distributions $\mu$ and $\nu$ introduced in [3] is defined as
$$d_{\mathcal{F}_{nn}}(\mu, \nu) = \sup_{f \in \mathcal{F}_{nn}} |\mathbb{E}_{x \sim \mu} f(x) - \mathbb{E}_{x \sim \nu} f(x)|, \tag{3}$$

where $\mathcal{F}_{nn}$ is a class of neural networks. In this paper, given the domain $\mathcal{X} \subseteq \mathbb{R}^h$, we let $\mathcal{F}_{nn}$ be the following set of depth-$d$ neural networks of the form:

$$f \in \mathcal{F}_{nn} : \mathbf{x} \in \mathcal{X} \longmapsto \mathbf{w}_d^T \sigma_{d-1}\left(\mathbf{W}_{d-1}\sigma_{d-2}\left(\cdots \sigma_1(\mathbf{W}_1\mathbf{x})\right)\right), \tag{4}$$

where $\mathbf{W}_i, i = 1, 2, ..., d-1$ are parameter matrices, $\mathbf{w}_d$ is a parameter vector (so that the output of the neural network is a scalar), and each $\sigma_i$ denotes the entry-wise activation function of layer $i$ for $i = 1, 2, \ldots, d-1$, i.e., for an input $\mathbf{z} \in \mathbb{R}^t$, $\sigma_i(\mathbf{z}) := [\sigma_i(z_1), \sigma_i(z_2), ..., \sigma_i(z_t)]^T$.

Throughout this paper, we adopt the following two assumptions on the activation functions in (4).

**Assumption 1.** *All activation functions $\sigma_i(\cdot)$ for $i = 1, 2, ..., d-1$ satisfy*

- *$\sigma_i(\cdot)$ is continuous and non-decreasing and $\sigma_i(0) \geq 0$.*

- *$\sigma_i(\cdot)$ is $L_i$-Lipschitz, where $L_i > 0$.*

**Assumption 2.** *For all activation functions $\sigma_i$, $i = 1, 2, ..., d-1$, there exist positive constants $q(i)$ and $Q_\sigma(i)$ such that for any $0 \leq x_1 \leq x_2 \leq q(i)$, $\sigma_i(x_2) - \sigma_i(x_1) \geq Q_\sigma(i)(x_2 - x_1)$.*

Note that Assumptions 1 and 2 hold for a variety of commonly used activation functions including ReLU, sigmoid, softPlus and tanh. In particular, in Assumption 2, the existence of the constants $q(i)$ and $Q_\sigma(i)$ are more important than the particular values they take, which affect only the constant terms in our bounds presented later. For example, Assumption 2 holds for ReLU for any $q(i) \leq \infty$ and $Q_\sigma(i) = 1$, and holds for sigmoid for any $q(i) > 0$ and $Q_\sigma(i) = 1/(2 + 2e^{q(i)})$.

As shown in [2], the practical training of GANs is conducted over neural networks with parameters lying in a compact space. Thus, we consider the following two compact parameter sets as taken in [5, 8, 22, 24],

$$\mathcal{W}_{1,\infty} := \prod_{i=1}^{d-1} \left\{\mathbf{W}_i \in \mathbb{R}^{n_i \times n_{i+1}} : \|\mathbf{W}_i\|_{1,\infty} \leq M_{1,\infty}(i)\right\} \times \left\{\mathbf{w}_d \in \mathbb{R}^{n_d} : \|\mathbf{w}_d\|_1 \leq M_{1,\infty}(d)\right\},$$

$$\mathcal{W}_F := \prod_{i=1}^{d-1} \left\{\mathbf{W}_i \in \mathbb{R}^{n_i \times n_{i+1}} : \|\mathbf{W}_i\|_F \leq M_F(i)\right\} \times \left\{\mathbf{w}_d \in \mathbb{R}^{n_d} : \|\mathbf{w}_d\| \leq M_F(d)\right\}. \tag{5}$$

## 2.2 Minimax Estimation Problem

In this paper, we study the minimax estimation problem defined as follows. Supposed $\mathcal{P}$ is a subset of Borel probability measures of interest. Let $\hat{d}(n, m)$ denote any estimator of the neural net distance $d_{\mathcal{F}_{nn}}(\mu, \nu)$ constructed by using the samples $\{\mathbf{x}_i\}_{i=1}^n$ and $\{\mathbf{y}_j\}_{j=1}^m$ respectively generated i.i.d. by $\mu, \nu \in \mathcal{P}$. Our goal is to first find a lower bound $C_l(\mathcal{P}, n, m)$ on the estimation error such that

$$\inf_{\hat{d}(n,m)} \sup_{\mu,\nu \in \mathcal{P}} \mathbb{P}\left\{|d_{\mathcal{F}_{nn}}(\mu, \nu) - \hat{d}(n, m)| \geq C_l(\mathcal{P}, n, m)\right\} > 0, \tag{6}$$

where $\mathbb{P}$ is the probability measure with respect to the random samples $\{\mathbf{x}_i\}_{i=1}^n$ and $\{\mathbf{y}_i\}_{i=1}^m$. We then focus on the empirical estimator $d_{\mathcal{F}_{nn}}(\hat{\mu}_n, \hat{\nu}_m)$ and are interested in finding an upper bound $C_u(\mathcal{P}, n, m)$ on the estimation error such that for any arbitrarily small $\delta > 0$,

$$\sup_{\mu,\nu \in \mathcal{P}} \mathbb{P}\left\{|d_{\mathcal{F}_{nn}}(\mu, \nu) - d_{\mathcal{F}_{nn}}(\hat{\mu}_n, \hat{\nu}_m)| \leq C_u(\mathcal{P}, n, m)\right\} > 1 - \delta. \tag{7}$$

Clearly such an upper bound also holds if the left hand side of (7) is defined in the same minimax sense as in (6).

It can be seen that the minimax estimation problem is defined with respect to the set $\mathcal{P}$ of distributions that $\mu$ and $\nu$ belong to. In this paper, we consider the set of all sub-Gaussian distributions over $\mathbb{R}^h$. We further divide the set into the two subsets and analyze them separately, for which the technical tools are very different. The first set $\mathcal{P}_{\mathrm{uB}}$ contains all sub-Gaussian distributions with *unbounded* support, and *bounded* mean and variance. Specifically, we assume that there exist $\tau > 0$ and $\Gamma_{\mathrm{uB}} > 0$ such that for any probability measure $\mu \in \mathcal{P}_{\mathrm{uB}}$ and any vector $\mathbf{a} \in \mathbb{R}^h$,

$$\mathbb{E}_{\mathbf{x} \sim \mu} e^{\mathbf{a}^T(\mathbf{x} - \mathbb{E}(\mathbf{x}))} \leq e^{\|\mathbf{a}\|^2 \tau^2 / 2} \text{ with } 0 < \tau, \|\mathbb{E}(\mathbf{x})\| \leq \Gamma_{\mathrm{uB}}. \tag{8}$$

The second class $\mathcal{P}_{\mathrm{B}}$ of distributions contains all sub-Gaussian distributions with bounded support $\mathcal{X} = \{\mathbf{x} : \|\mathbf{x}\| \leq \Gamma_{\mathrm{B}}\} \subset \mathbb{R}^h$ (note that this set in fact includes all distributions with bounded support). These two mutually exclusive classes cover most probability distributions in practice.

# 3 Main Results

## 3.1 Minimax Lower Bound

We first develop the following minimax lower bound for the sub-Gaussian distribution class $\mathcal{P}_{uB}$ with unbounded support.

**Theorem 1 (unbounded-support sub-Gaussian class $\mathcal{P}_{uB}$).** *Let $\mathcal{F}_{nn}$ be the set of neural networks defined by (4). For the parameter set $\mathcal{W}_F$ in (5), if $\sqrt{m^{-1} + n^{-1}} < \sqrt{3}q(1)/(2M_F(1)\Gamma_{uB})$, then*

$$\inf_{\hat{d}(n,m)} \sup_{\mu,\nu \in \mathcal{P}_{uB}} \mathbb{P}\left\{\left|d_{\mathcal{F}_{nn}}(\mu,\nu) - \hat{d}(n,m)\right| \geq C(\mathcal{P}_{uB})\max\left(n^{-1/2},\, m^{-1/2}\right)\right\} \geq \frac{1}{4}, \quad (9)$$

*where*

$$C(\mathcal{P}_{uB}) = \frac{\sqrt{3}}{6}M_F(1)M_F(d)\Gamma_{uB}\left(1 - \Phi\left(\frac{q(1)}{2M_F(1)\Gamma_{uB}}\right)\right)\prod_{i=2}^{d-1}\Omega(i)\prod_{i=1}^{d-1}Q_\sigma(i), \quad (10)$$

*and $\Phi(\cdot)$ is the cumulative distribution function (CDF) of the standard Gaussian distribution and the constants $\Omega(i), i = 2, 3, ..., d-1$ are given by the following recursion*

$$\Omega(2) = \min\left\{M_F(2),\, q(2)\big/\sigma_1(q(1))\right\},$$
$$\Omega(i) = \min\left\{M_F(i),\, q(i)\big/\sigma_{i-1}(\Omega(i-1)\cdots\Omega(2)\sigma_1(q(1)))\right\} \text{ for } i = 3, 4, ..., d-1. \quad (11)$$

*The same result holds for the parameter set $\mathcal{W}_{1,\infty}$ by replacing $M_F(i)$ in (10) with $M_{1,\infty}(i)$.*

Theorem 1 implies that $d_{\mathcal{F}_{nn}}(\mu,\nu)$ cannot be estimated at a rate faster than $\max\left(n^{-1/2},\, m^{-1/2}\right)$ by any estimator over the class $\mathcal{P}_{uB}$. The proof of Theorem 1 is based on the Le Cam's method. Such a technique was also used in [26] to derive the minimax lower bound for estimating MMD. However, our technical development is quite different from that in [26]. In specific, one major step of the Le Cam's method is to lower-bound the difference arising due to two hypothesis distributions. In the MMD case in [26], MMD can be expressed in a closed form for the chosen distributions. Hence, the lower bound in Le Cam's method can be derived based on such a closed form of MMD. As a comparison, the neural net distance does not have a closed-form expression for the chosen distributions. As a result, our derivation involves lower-bounding the difference of the expectations of the neural network function with respect to two corresponding distributions. Such developments require substantial machineries to deal with the complicated multi-layer structure of neural networks. See Appendix A.1 for more details.

For general neural networks, $C(\mathcal{P}_{uB})$ takes a complicated form as in (10). We next specialize to ReLU networks to illustrate how this constant depends on the neural network parameters.

**Corollary 1.** *Under the setting of Theorem 1, suppose each activation function is ReLU, i.e., $\sigma_i(z) = \max\{0, z\}$, $i = 1, 2, ..., d-1$. For the parameter set $\mathcal{W}_F$ and all $m, n \geq 1$, we have*

$$\inf_{\hat{d}(n,m)} \sup_{\mu,\nu \in \mathcal{P}_{uB}} \mathbb{P}\left\{\left|d_{\mathcal{F}_{nn}}(\mu,\nu) - \hat{d}(n,m)\right| \geq 0.08\,\Gamma_{uB}\prod_{i=1}^{d}M_F(i)\max\left(n^{-1/2}, m^{-1/2}\right)\right\} \geq \frac{1}{4}.$$

*The same result holds for the parameter set $\mathcal{W}_{1,\infty}$ by replacing $M_F(i)$ with $M_{1,\infty}(i)$.*

Next, we provide the minimax lower bound for the distribution class $\mathcal{P}_B$ with bounded support. The proof (see Appendix B) is also based on the Le Cam's method, but with the construction of distributions having the bounded support sets, which are different from those for Theorem 1.

**Theorem 2 (bounded-support class $\mathcal{P}_B$).** *Let $\mathcal{F}_{nn}$ be the set of neural networks defined by (4). For the parameter set $\mathcal{W}_F$, we have*

$$\inf_{\hat{d}(n,m)} \sup_{\mu,\nu \in \mathcal{P}_B} \mathbb{P}\left\{\left|d_{\mathcal{F}_{nn}}(\mu,\nu) - \hat{d}(n,m)\right| \geq C(\mathcal{P}_B)\max\left(n^{-1/2}, m^{-1/2}\right)\right\} \geq \frac{1}{4}, \quad (12)$$

*where*

$$C(\mathcal{P}_B) = 0.17\left(M_F(d)\sigma_{d-1}(\cdots\sigma_1(M_F(1)\Gamma_B)) - M_F(d)\sigma_{d-1}(\cdots\sigma_1(-M_F(1)\Gamma_B))\right), \quad (13)$$

*where all constants $M_F(i), i = 1, 2, ..., d$ in the second term of the right side of (13) have negative signs. The same result holds for the parameter set $\mathcal{W}_{1,\infty}$ by replacing $M_F(i)$ in (13) with $M_{1,\infty}(i)$.*

**Corollary 2.** *Under the setting of Theorem 2, suppose that each activation function is ReLU. For the parameter set $\mathcal{W}_F$, we have,*

$$\inf_{\hat{d}(n,m)} \sup_{\mu,\nu \in \mathcal{P}_{\mathrm{B}}} \mathbb{P} \left\{ \left| d_{\mathcal{F}_{nn}}(\mu,\nu) - \hat{d}(n,m) \right| \geq 0.17 \, \Gamma_{\mathrm{B}} \prod_{i=1}^{d} M_F(i) \max\left( n^{-1/2}, \, m^{-1/2} \right) \right\} \geq \frac{1}{4}.$$

*The same result holds for the parameter set $\mathcal{W}_{1,\infty}$ by replacing $M_F(i)$ with $M_{1,\infty}(i)$.*

### 3.2 Rademacher Complexity-based Upper Bound

In this subsection, we provide an upper bound on $|d_{\mathcal{F}_{nn}}(\mu,\nu) - d_{\mathcal{F}_{nn}}(\hat{\mu}_n, \hat{\nu}_m)|$, which serves as an upper bound on the minimax estimation error. Our main technical development lies in deriving the bound for the unbounded-support sub-Gaussian class $\mathcal{P}_{\mathrm{uB}}$, which requires a number of new technical developments. We discuss its proof in Section 3.4.

**Theorem 3** (**unbounded-support sub-Gaussian class $\mathcal{P}_{\mathbf{uB}}$**). *Let $\mathcal{F}_{nn}$ be the set of neural networks defined by (4), and suppose that two distributions $\mu, \nu \in \mathcal{P}_{\mathrm{uB}}$ and $\hat{\mu}_n, \hat{\nu}_m$ are their empirical measures. For a constant $\delta > 0$ satisfying $\sqrt{6h} \min\{n,m\}\sqrt{m^{-1} + n^{-1}} \geq 4\sqrt{\log(1/\delta)}$, we have*

**(I)** *If the parameter set is $\mathcal{W}_F$ and each activation function satisfies $\sigma_i(\alpha x) = \alpha \sigma_i(x)$ for all $\alpha > 0$ (e.g., ReLU or leaky ReLU), then with probability at least $1 - \delta$ over the randomness of $\hat{\mu}_n$ and $\hat{\nu}_m$,*

$$|d_{\mathcal{F}_{nn}}(\mu,\nu) - d_{\mathcal{F}_{nn}}(\hat{\mu}_n, \hat{\nu}_m)|$$
$$\leq 2\Gamma_{\mathrm{uB}} \prod_{i=1}^{d} M_F(i) \prod_{i=1}^{d-1} L_i \left( \sqrt{6d\log 2 + 5h/4} + \sqrt{2h\log(1/\delta)} \right) \left( n^{-1/2} + m^{-1/2} \right).$$

**(II)** *If the parameter set is $\mathcal{W}_{1,\infty}$ and each activation function satisfies $\sigma_i(0) = 0$ (e.g., ReLU, leaky ReLU or tanh), then with probability at least $1 - \delta$ over the randomness of $\hat{\mu}_n$ and $\hat{\nu}_m$,*

$$|d_{\mathcal{F}_{nn}}(\mu,\nu) - d_{\mathcal{F}_{nn}}(\hat{\mu}_n, \hat{\nu}_m)|$$
$$\leq 2\Gamma_{\mathrm{uB}} \prod_{i=1}^{d} M_{1,\infty}(i) \prod_{i=1}^{d-1} L_i \left( \sqrt{2d\log 2 + 2\log h} + \sqrt{2h\log(1/\delta)} \right) \left( n^{-1/2} + m^{-1/2} \right).$$

**Corollary 3.** *Theorem 3 is directly applicable to ReLU networks with $L_i = 1$ for $i = 1, \ldots, d$.*

We next present an upper bound on $|d_{\mathcal{F}_{nn}}(\mu,\nu) - d_{\mathcal{F}_{nn}}(\hat{\mu}_n, \hat{\nu}_m)|$ for the bounded-support class $\mathcal{P}_{\mathrm{B}}$. In such a case, each data sample $\mathbf{x}_i$ satisfies $\|\mathbf{x}_i\| \leq \Gamma_{\mathrm{B}}$, and hence we apply the standard McDiarmid inequality [16] and the Rademacher complexity bounds in [8] to upper-bound $|d_{\mathcal{F}_{nn}}(\mu,\nu) - d_{\mathcal{F}_{nn}}(\hat{\mu}_n, \hat{\nu}_m)|$. The detailed proof can be found in Appendix D.

**Theorem 4** (**bounded-support class $\mathcal{P}_{\mathbf{B}}$**). *Let $\mathcal{F}_{nn}$ be the set of neural networks defined by (4), and suppose that two distributions $\mu, \nu \in \mathcal{P}_{\mathrm{B}}$. Then, we have*

**(I)** *If the parameter set is $\mathcal{W}_F$ and each activation function satisfies $\sigma_i(\alpha x) = \alpha \sigma_i(x)$ for all $\alpha > 0$, then with probability at least $1 - \delta$ over the randomness of $\hat{\mu}_n$ and $\hat{\nu}_m$,*

$$|d_{\mathcal{F}_{nn}}(\mu,\nu) - d_{\mathcal{F}_{nn}}(\hat{\mu}_n, \hat{\nu}_m)|$$
$$\leq \sqrt{2}\Gamma_{\mathrm{B}} \prod_{i=1}^{d} M_{1,\infty}(i) \prod_{i=1}^{d-1} L_i \left( 2\sqrt{d\log 2} + \sqrt{\log(1/\delta)} + \sqrt{2} \right) \left( n^{-1/2} + m^{-1/2} \right).$$

**(II)** *If the parameter set is $\mathcal{W}_{1,\infty}$ and each activation function satisfies $\sigma_i(0) = 0$, then with probability at least $1 - \delta$ over the randomness of $\hat{\mu}_n$ and $\hat{\nu}_m$,*

$$|d_{\mathcal{F}_{nn}}(\mu,\nu) - d_{\mathcal{F}_{nn}}(\hat{\mu}_n, \hat{\nu}_m)|$$
$$\leq \Gamma_{\mathrm{B}} \prod_{i=1}^{d} M_{1,\infty}(i) \prod_{i=1}^{d-1} L_i \left( 4\sqrt{d + 1 + \log h} + \sqrt{2\log(1/\delta)} \right) \left( n^{-1/2} + m^{-1/2} \right),$$

**Corollary 4.** *Theorem 4 is applicable for ReLU networks with $L_i = 1$ for $i = 1, \ldots, d$.*

As a comparison, the upper bound derived in [3] is linear with the total number of the parameters of neural networks, whereas our bound in Theorem 4 scales only with the square root of depth $\sqrt{d}$ (and other terms in Theorem 4 matches the lower bound in Corollary 2), which is much smaller.

## 3.3 Optimality of Minimax Estimation and Discussions

We compare the lower and upper bounds and make the following remarks on the optimality of minimax estimation of the neural net distance.

- For the unbounded-support sub-Gaussian class $\mathcal{P}_{\text{uB}}$, comparison of Theorems 1 and 3 indicates that the empirical estimator $d_{\mathcal{F}_{nn}}(\hat{\mu}_n, \hat{\nu}_m)$ achieves the optimal minimax estimation rate $\max\{n^{-1/2}, m^{-1/2}\}$ as the sample size goes to infinity.

- Furthermore, for ReLU networks, comparison of Corollaries 1 and 3 implies that the lower and upper bounds match further in terms of $\Gamma_{\text{uB}} \prod_{i=1}^{d} M(i) \max\left\{n^{-1/2}, m^{-1/2}\right\}$, where $M(i)$ can be $M_F(i)$ or $M_{1,\infty}(i)$, indicating that both $\prod_{i=1}^{d} \|\mathbf{W}_i\|_F$ and $\prod_{i=1}^{d} \|\mathbf{W}_i\|_{1,\infty}$ capture the estimation accuracy. Such an observation is consistent with those made in [20] for the generalization error of training deep neural networks. Moreover, the mean norm $\|\mathbb{E}(\mathbf{x})\|$ and the variance parameter of the distributions also determine the estimation accuracy due to the match of the bounds in $\Gamma_{\text{uB}}$.

- The same observations hold for the bounded-support class $\mathcal{P}_{\text{B}}$ by comparing Theorems 2 and 4 as well as comparing Corollaries 2 and 4.

We further note that for ReLU networks, for both the unbounded-support sub-Gaussian class $\mathcal{P}_{\text{uB}}$ and the bounded-support class $\mathcal{P}_{\text{B}}$, there is a gap of $\sqrt{d}$ (or $\sqrt{d+h}$, $\sqrt{d + \log h}$ depending on the distribution class and the norm of the weight matrices). To close the gap, the size-independent bound on Rademacher complexity in [8] appears appealing. However, such a bound is applicable only to the bounded-support class $\mathcal{P}_{\text{B}}$, and helps to remove the dependence on $\sqrt{d}$ but at the cost of sacrificing the rate (i.e., from $m^{-1/2} + n^{-1/2}$ to $m^{-1/4} + n^{-1/4}$). Consequently, such an upper bound matches the lower bound in Corollary 2 for ReLU networks over the network parameters, but not in terms of the sample size, and is interesting only in the regime when $d \gg \max\{n, m\}$. It is thus still an open problem and calling for future efforts to close the gap of $\sqrt{d}$ for estimating the neural net distance.

## 3.4 Proof Outline for Theorem 3

In this subsection, we briefly explain the three major steps to prove Theorem 3, because some of these intermediate steps correspond to theorems that can be of independent interest. The detailed proof can be found in Appendix C.

**Step 1: A new McDiarmid's type of inequality.** To establish an upper bound on $|d_{\mathcal{F}_{nn}}(\mu, \nu) - d_{\mathcal{F}_{nn}}(\hat{\mu}_n, \hat{\nu}_m)|$, the standard McDiarmid's inequality [16] that requires the bounded difference condition is not applicable here, because the input data has unbounded support so that the functions in $\mathcal{F}_{nn}$ can be unbounded, e.g., ReLU neural networks. Such a challenge can be addressed by a generalized McDiarmid's inequality for *scalar* sub-Gaussian variables established in [11]. However, the input data are vectors in our setting. Thus, we further generalize the result in [11] and establish the following new McDiarmid's type of concentration inequality for *unbounded sub-Gaussian* random *vectors* and Lipschitz (*possibly unbounded*) functions. Such development turns out to be nontrivial, which requires further machineries and tail bound inequalities (see detailed proof in Appendix C.1).

**Theorem 5.** *Let* $\{\mathbf{x}_i\}_{i=1}^{n} \overset{i.i.d.}{\sim} \mu$ *and* $\{\mathbf{y}_i\}_{i=1}^{m} \overset{i.i.d.}{\sim} \nu$ *be two collections of random variables, where* $\mu, \nu \in \mathcal{P}_{\text{uB}}$ *are two unbounded-support sub-Gaussian distributions over* $\mathbb{R}^h$. *Suppose that* $F : (\mathbb{R}^h)^{n+m} \longmapsto \mathbb{R}$ *is a function of* $\mathbf{x}_1, ..., \mathbf{x}_n, \mathbf{y}_1, ..., \mathbf{y}_m$, *which satisfies for any* $i$,

$$|F(\mathbf{x}_1, ..., \mathbf{x}_i, ...., \mathbf{y}_m) - F(\mathbf{x}_1, ..., \mathbf{x}_i', ...., \mathbf{y}_m)| \leq L_{\mathcal{F}} \|\mathbf{x}_i - \mathbf{x}_i'\|/n,$$

$$|F(\mathbf{x}_1, ..., \mathbf{y}_i, ...., \mathbf{y}_m) - F(\mathbf{x}_1, ..., \mathbf{y}_i', ...., \mathbf{y}_m)| \leq L_{\mathcal{F}} \|\mathbf{y}_i - \mathbf{y}_i'\|/m. \quad (14)$$

*Then, for all* $0 \leq \epsilon \leq \sqrt{3}h\Gamma_{\text{uB}}L_{\mathcal{F}} \min\{m, n\}(n^{-1} + m^{-1})$,

$$\mathbb{P}\left(F(\mathbf{x}_1, ..., \mathbf{x}_n, ...., \mathbf{y}_m) - \mathbb{E} F(\mathbf{x}_1, ..., \mathbf{x}_n, ...., \mathbf{y}_m) \geq \epsilon\right) \leq \exp\left(\frac{-\epsilon^2 mn}{8h\Gamma_{\text{uB}}^2 L_{\mathcal{F}}^2 (m+n)}\right). \quad (15)$$

**Step 2: Upper bound based on Rademacher complexity.** By applying Theorem 5, we derive an upper bound on $|d_{\mathcal{F}_{nn}}(\mu, \nu) - d_{\mathcal{F}_{nn}}(\hat{\mu}_n, \hat{\nu}_m)|$ in terms of the average Rademacher complexity that we define below.

**Definition 1.** *The average Rademacher complexity $\mathcal{R}_n(\mathcal{F}_{nn}, \mu)$ corresponding to the distribution $\mu$ with $n$ samples is defined as $\mathcal{R}_n(\mathcal{F}_{nn}, \mu) = \mathbb{E}_{\mathbf{x},\epsilon} \sup_{f \in \mathcal{F}_{nn}} \left| \frac{1}{n} \sum_{i=1}^n \epsilon_i f(\mathbf{x}_i) \right|$, where $\{\mathbf{x}_i\}_{i=1}^n$ are generated i.i.d. by $\mu$ and $\{\epsilon_i\}_{i=1}^n$ are independent random variables chosen from $\{-1, 1\}$ uniformly.*

Then, we have the following result with the proof provided in Appendix C.2. Recall that $L_i$ is the Lipchitz constant of the activation function $\sigma_i(\cdot)$.

**Theorem 6.** *Let $\mathcal{F}_{nn}$ be the set of neural networks defined by (4). For the parameter set $\mathcal{W}_F$ defined in (5), suppose that $\mu, \nu \in \mathcal{P}_{\text{uB}}$ are two sub-Gaussian distributions satisfying (8) and $\hat{\mu}_n, \hat{\nu}_m$ are the empirical measures of $\mu, \nu$. If $\sqrt{6h} \min\{n, m\}\sqrt{m^{-1} + n^{-1}} \geq 4\sqrt{\log(1/\delta)}$, then with probability at least $1 - \delta$ over the randomness of $\hat{\mu}_n$ and $\hat{\nu}_m$,*

$$|d_{\mathcal{F}_{nn}}(\mu, \nu) - d_{\mathcal{F}_{nn}}(\hat{\mu}_n, \hat{\nu}_m)|$$

$$\leq 2\mathcal{R}_n(\mathcal{F}_{nn}, \mu) + 2\mathcal{R}_m(\mathcal{F}_{nn}, \nu) + 2\Gamma_{\text{uB}} \prod_{i=1}^d M_F(i) \prod_{i=1}^{d-1} L_i \sqrt{2h\left(n^{-1} + m^{-1}\right)\log(1/\delta)}. \quad (16)$$

*The same result holds for the parameter set $\mathcal{W}_{1,\infty}$ by replacing $M_F(i)$ in (16) with $M_{1,\infty}(i)$.*

**Step 3: Average Rademacher complexity bound for unbounded sub-Gaussian variables.** We derive an upper bound on the Rademacher complexity $\mathcal{R}_n(\mathcal{F}_{nn}, \mu)$. In particular, as we explain next, our upper bound is tighter than directly applying the existing bounds in [8, 22]. To see this, [8, 22] provided upper bounds on the data-dependent Rademacher complexity of neural networks defined by $\hat{\mathcal{R}}_n(\mathcal{F}_{nn}) = \mathbb{E}_\epsilon \sup_{f \in \mathcal{F}_{nn}} \frac{1}{n} \sum_{i=1}^n \epsilon_i f(\mathbf{x}_i)$. For the parameter set $\mathcal{W}_F$, [22] showed that $\hat{\mathcal{R}}_n(\mathcal{F}_{nn})$ was bounded by $2^d \prod_{i=1}^d M_F(i) \prod_{i=1}^{d-1} L_i \sqrt{\sum_{i=1}^n \|\mathbf{x}_i\|^2}/n$, and [8] further improved this bound to $(\sqrt{2\log(2)d} + 1) \prod_{i=1}^d M_F(i) \prod_{i=1}^{d-1} L_i \sqrt{\sum_{i=1}^n \|\mathbf{x}_i\|^2}/n$. Directly applying this result for unbounded sub-Gaussian inputs $\{\mathbf{x}_i\}_{i=1}^n$ yields

$$\mathbb{E}_{\mathbf{x}} \hat{\mathcal{R}}_n(\mathcal{F}_{nn}) \leq \mathcal{O}\left( \Gamma_{\text{uB}} \prod_{i=1}^d M_F(i) \prod_{i=1}^{d-1} L_i \sqrt{dh}/\sqrt{n} \right). \quad (17)$$

We next show that by exploiting the sub-Gaussianity of the input data, we provide an improved bound on the average Rademacher complexity. The detailed proof can be found in Appendix C.3.

**Theorem 7.** *Let $\mathcal{F}_{nn}$ be the set of neural networks defined by (4), and let $\mathbf{x}_1, ..., \mathbf{x}_n \in \mathbb{R}^h$ be i.i.d. random samples generated by an unbounded-supported sub-Gaussian distribution $\mu \in \mathcal{P}_{\text{uB}}$. Then,*

**(I)** *If the parameter set is $\mathcal{W}_F$ and activation functions satisfy $\sigma_i(\alpha x) = \alpha \sigma_i(x)$ for all $\alpha > 0$, then*

$$\mathcal{R}_n(\mathcal{F}_{nn}, \mu) \leq \Gamma_{\text{uB}} \prod_{i=1}^d M_F(i) \prod_{i=1}^{d-1} L_i \sqrt{6d\log 2 + 5h/4}/\sqrt{n}. \quad (18)$$

**(II)** *If the parameter set is $\mathcal{W}_{1,\infty}$ and each activation function satisfies $\sigma_i(0) = 0$, then*

$$\mathcal{R}_n(\mathcal{F}_{nn}, \mu) \leq \sqrt{2}\Gamma_{\text{uB}} \prod_{i=1}^d M_{1,\infty}(i) \prod_{i=1}^{d-1} L_i \sqrt{d\log 2 + \log h}/\sqrt{n}. \quad (19)$$

Theorem 7 indicates that for the parameter set $\mathcal{W}_F$, our upper bound in (18) replaces the order dependence $\mathcal{O}(\sqrt{dh})$ in (17) to $\mathcal{O}(\sqrt{d+h})$, and hence our proof has the order-level improvement than directly using the existing bounds. The same observation can be made for the parameter set $\mathcal{W}_{1,\infty}$. Such improvement is because our proof takes advantage of the sub-Gaussianity of the inputs whereas the bounds in [8, 22] must hold for any data input (and hence the worst-case data input).

We also note that [23] provided an upper bound on the Rademacher complexity for one-hidden-layer neural networks for Gaussian inputs. Casting Lemma 3.2 in [23] to our setting of (18) yields

$$\mathcal{R}_n(\mathcal{F}_{nn}, \mu) \leq \mathcal{O}\left( \Gamma_{\text{uB}} M_F(2) M_F(1) L_1 \sqrt{n_1 h}/\sqrt{n} \right), \quad (20)$$

where $n_1$ is the number of neurons in the hidden layer. Compared with (20), our bound has an order-level $\mathcal{O}(\sqrt{n_1})$ improvement.

**Summary.** Therefore, Theorem 3 follows by combining Theorems 6 and 7 and using the fact that $\sqrt{1/n + 1/m} < \sqrt{1/n} + \sqrt{1/m}$.

# 4 Conclusion

In this paper, we developed both the lower and upper bounds for the minimax estimation of the neural net distance based on finite samples. Our results established the minimax optimality of the empirical estimator in terms of not only the sample size but also the norm of the parameter matrices of neural networks, which justifies its usage for training GANs.

**Acknowledgments**

The work was supported in part by U.S. National Science Foundation under the grant CCF-1801855.

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
