[Supplementary Material]

# A Proofs of Theorem 1 and Corollary 1

## A.1 Proof of Theorem 1

The proof is based on a variant of the Le Cam's method in Theorem 3 in [26]. Our major technical developments lie in properly choosing two hypothesis distributions as well as a neural network, and then lower-bounding the difference of the expectation of the chosen neural network function between the two distributions. We first present the Le Cam's method (Theorem 3 in [26]).

**Lemma 1** (Le Cam's method). *Let $F : \Theta \to \mathbb{R}$ be a functional defined on a space $\Theta$ and $P_\Theta = \{P_\theta : \theta \in \Theta\}$ be a set of probability measures. The data samples $\mathcal{D}_n$ are distributed according to an unknown element $P_\theta \in P_\Theta$. Assume that there exist $\theta_1, \theta_2 \in \Theta$ such that $|F(\theta_1) - F(\theta_2)| \geq 2\beta > 0$ and $\mathrm{KL}(P_{\theta_2} \| P_{\theta_1}) \leq \alpha < \infty$. Then,*

$$\inf_{\hat{F}_n} \sup_{\theta \in \Theta} P_\theta \left\{ \left| \hat{F}_n(\mathcal{D}_n) - F(\theta) \right| \geq \beta \right\} \geq \max \left\{ \frac{1}{4} e^{-\alpha}, \frac{1 - \sqrt{\alpha/2}}{2} \right\}, \tag{21}$$

where the Kullback-Leibler divergence $\mathrm{KL}(P_{\theta_2} \| P_{\theta_1}) := \int \log \left( \frac{dP_{\theta_2}}{dP_{\theta_1}} \right) dP_{\theta_2}$ and $\hat{F}_n(\mathcal{D}_n)$ is an estimator of $F(\theta)$ based on the random samples $\mathcal{D}_n$.

We first consider the case when the parameter set is $\mathcal{W}_F$, and then adapt the proof to the parameter set $\mathcal{W}_{1,\infty}$. In addition, we suppose $m \geq n$, and the case $m < n$ can be proved in the same way.

**Case 1: parameter set $\mathcal{W}_F$.** Recall from (5) that $\mathcal{W}_F$ is defined by

$$\mathcal{W}_F := \prod_{i=1}^{d-1} \left\{ \mathbf{W}_i \in \mathbb{R}^{n_i \times m_i} : \|\mathbf{W}_i\|_F \leq M_F(i) \right\} \times \left\{ \mathbf{w}_d \in \mathbb{R}^{n_d} : \|\mathbf{w}_d\| \leq M_F(d) \right\}.$$

Assign each distribution $\mu \in \mathcal{P}_{\mathrm{uB}}$ with a unique index $\tilde{\theta}_\mu$ and define an index set $\tilde{\Theta} := \{\tilde{\theta}_\mu : \mu \in \mathcal{P}_{\mathrm{uB}}\}$. To cast the form (9) in Theorem 1 to the context of Lemma 1, we let $\Theta = \tilde{\Theta} \times \tilde{\Theta}$, $F(\theta) = d_{\mathcal{F}_{nn}}(\mu, \nu)$ and $P_\theta := \mathbb{P} = \mu^n \times \nu^m$ in (21) for $\theta = (\tilde{\theta}_\mu, \tilde{\theta}_\nu)$, where $\tilde{\theta}_\mu$ and $\tilde{\theta}_\nu$ are the indices of $\mu, \nu \in \mathcal{P}_{\mathrm{uB}}$ and $\mathbb{P} = \mu^n \times \nu^m$ is the probability measure with the respect to the random samples $\{\mathbf{x}_i\}_{i=1}^n \overset{i.i.d.}{\sim} \mu$ and $\{\mathbf{y}_i\}_{i=1}^m \overset{i.i.d.}{\sim} \nu$.

To apply Lemma 1, we need to find two pairs of distributions $(\mu_1, \nu_1), (\mu_2, \nu_2) \in \mathcal{P}_{\mathrm{uB}}$ and $\alpha, \beta$ such that $|d_{\mathcal{F}_{nn}}(\mu_1, \nu_1) - d_{\mathcal{F}_{nn}}(\mu_2, \nu_2)| \geq 2\beta$ and $\mathrm{KL}(\mathbb{P}_2 \| \mathbb{P}_1) \leq \alpha$, where $\mathbb{P}_1 = \mu_1^n \times \nu_1^m$ and $\mathbb{P}_2 = \mu_2^n \times \nu_2^m$. Specifically, we choose the following four Gaussian distributions

$$\mu_1 = G(\mathbf{u}_1, \tau^2 \mathbf{I}_h), \quad \nu_1 = G(\mathbf{u}_2, \tau^2 \mathbf{I}_h), \quad \mu_2 = \nu_2 = G(0, \tau^2 \mathbf{I}_h). \tag{22}$$

where

$$\|\mathbf{u}_1\|^2 = \frac{\Gamma_{\mathrm{uB}}^2}{3} \left( \frac{1}{n} + \frac{1}{m} \right), \|\mathbf{u}_2\|^2 = \frac{\Gamma_{\mathrm{uB}}^2}{3m}, \mathbf{u}_1^T \mathbf{u}_2 = \|\mathbf{u}_2\|^2, \tau^2 = \frac{\Gamma_{\mathrm{uB}}^2}{3} \left( 2 + \frac{n}{m} \right) \tag{23}$$

with $\Gamma_{\mathrm{uB}}$ defined as the upper bound of the mean and variance parameter of the unbounded-support sub-Gaussian distributions in $\mathcal{P}_{\mathrm{uB}}$ (see (8) for the definition). Clearly, (23) implies that $\|\mathbf{u}_1 - \mathbf{u}_2\|^2 = \Gamma_{\mathrm{uB}}^2/3n$ and $0 \leq \tau, \|\mathbf{u}_1\|, \|\mathbf{u}_2\| \leq \Gamma_{\mathrm{uB}}$.

Since $\mu_2 = \nu_2$, $d_{\mathcal{F}_{nn}}(\mu_2, \nu_2) = 0$, and hence

$$|d_{\mathcal{F}_{nn}}(\mu_1, \nu_1) - d_{\mathcal{F}_{nn}}(\mu_2, \nu_2)| = d_{\mathcal{F}_{nn}}(\mu_1, \nu_1) = \sup_{f \in \mathcal{F}_{nn}} |\mathbb{E}_{\mathbf{x} \sim \mu_1} f(\mathbf{x}) - \mathbb{E}_{\mathbf{x} \sim \nu_1} f(\mathbf{x})|. \tag{24}$$

To lower-bound (24), we choose the weights in $\tilde{f}(\mathbf{x}) = \tilde{\mathbf{w}}_d^T \sigma_{d-1}(\tilde{\mathbf{W}}_{d-1} \sigma_{d-2}(\cdots \sigma_1(\tilde{\mathbf{W}}_1 \mathbf{x}))) \in \mathcal{F}_{nn}$ as follows:

    1. $\tilde{\mathbf{w}}_d(1) = M_F(d), \tilde{\mathbf{w}}_d(i) = 0$ for $i = 2, 3, ..., n_d$,

    2. For $i = 2, ..., d-1$, $\tilde{\mathbf{W}}_i(1,1) = \Omega(i), \tilde{\mathbf{W}}_i(s,t) = 0$ for $(s,t) \neq (1,1)$,

    3. $\|\tilde{\mathbf{W}}_1(1)\| = \mathbf{w}_1 = M_F(1)(\mathbf{u}_1 - \mathbf{u}_2)/\|\mathbf{u}_1 - \mathbf{u}_2\|, \tilde{\mathbf{W}}_1(s) = \mathbf{0}$ for $2 \leq s \leq n_1$,    (25)

where $\widetilde{\mathbf{w}}_d(i)$ refers to the $i^{th}$ coordinate of $\widetilde{\mathbf{w}}_d$, $\widetilde{\mathbf{W}}_i(s,t)$ denotes the $(s,t)^{th}$ entry of $\widetilde{\mathbf{W}}$, $\widetilde{\mathbf{W}}_1(s)$ is the $s^{th}$ column vector of $\widetilde{\mathbf{W}}_1^T$ and $\Omega(i)$ is defined in (11). Then, corresponding to the above parameters, we have

$$\tilde{f}(\mathbf{x}) = M_F(d)\sigma_{d-1}\left(\Omega(d-1)\sigma_{d-2}\cdots\Omega(2)\sigma_1\left(\mathbf{w}_1^T\mathbf{x}\right)\right). \tag{26}$$

Combining (24) and (26) yields

$$|d_{\mathcal{F}_{nn}}(\mu_1,\nu_1) - d_{\mathcal{F}_{nn}}(\mu_2,\nu_2)| = \sup_{f\in\mathcal{F}_{nn}} |\mathbb{E}_{\mathbf{x}\sim\mu_1}f(\mathbf{x}) - \mathbb{E}_{\mathbf{x}\sim\nu_1}f(\mathbf{x})|$$

$$\geq \left|\mathbb{E}_{\mathbf{x}\sim\mu_1}\tilde{f}(\mathbf{x}) - \mathbb{E}_{\mathbf{x}\sim\nu_1}\tilde{f}(\mathbf{x})\right|. \tag{27}$$

Due to the definitions of $\mu_1$ in (22), for $\mathbf{x} \sim \mu_1$, we have that $\mathbf{w}_1^T\mathbf{x} \sim G(\mathbf{w}_1^T\mathbf{u}_1, \|\mathbf{w}_1\|^2\tau^2)$. Let $x = \mathbf{w}_1^T\mathbf{x} \in \mathbb{R}$ and $\varphi(x\,;\,u,\tau^2)$ be the probability density of the Gaussian distribution with mean $u$ and variance $\tau^2$. Then, we have

$$\mathbb{E}_{\mathbf{x}\sim\mu_1}\tilde{f}(\mathbf{x}) = \int_x M_F(d)\sigma_{d-1}\left(\Omega(d-1)\sigma_{d-2}\cdots\Omega(2)\sigma_1(x)\right)\varphi\left(x\,;\,\mathbf{w}_1^T\mathbf{u}_1,\|\mathbf{w}_1\|^2\tau^2\right)\,\mathrm{d}x$$

$$= \int_x M_F(d)\sigma_{d-1}\left(\Omega(d-1)\sigma_{d-2}\cdots\sigma_1(x+\mathbf{w}_1^T\mathbf{u}_1)\right)\varphi\left(x\,;\,0,\|\mathbf{w}_1\|^2\tau^2\right)\,\mathrm{d}x. \tag{28}$$

Similarly, we have

$$\mathbb{E}_{\mathbf{x}\sim\nu_1}\tilde{f}(\mathbf{x}) = \int_x M_F(d)\sigma_{d-1}\left(\Omega(d-1)\sigma_{d-2}\cdots\Omega(2)\sigma_1(x)\right)\varphi\left(x\,;\,0,\|\mathbf{w}_1\|^2\tau^2\right)\,\mathrm{d}x,$$

which, in conjunction with (27) and (28), yields

$$|d_{\mathcal{F}_{nn}}(\mu_1,\nu_1) - d_{\mathcal{F}_{nn}}(\mu_2,\nu_2)| \geq \mathbb{E}_{\mathbf{x}\sim\mu_1}\tilde{f}(\mathbf{x}) - \mathbb{E}_{\mathbf{x}\sim\nu_1}\tilde{f}(\mathbf{x})$$

$$= \int_x \underbrace{\left(M_F(d)\sigma_{d-1}(\cdots\sigma_1(x+\mathbf{w}_1^T\mathbf{u}_1)) - M_F(d)\sigma_{d-1}(\cdots\sigma_1(x))\right)}_{\Delta(x)}\varphi\left(x\,;\,0,\|\mathbf{w}_1\|^2\tau^2\right)\,\mathrm{d}x. \tag{29}$$

Following from the definitions of $\mathbf{u}_1$ and $\mathbf{u}_2$ in (23), we have

$$\mathbf{w}_1^T\mathbf{u}_1 \overset{(i)}{\geq} \mathbf{w}_1^T\mathbf{u}_2 \overset{(ii)}{=} 0, \tag{30}$$

where (i) follows from the fact that $\mathbf{w}_1^T(\mathbf{u}_1 - \mathbf{u}_2) = M_F(1)\|\mathbf{u}_1 - \mathbf{u}_2\| \geq 0$ and (ii) follows because $\mathbf{u}_1^T\mathbf{u}_2 = \|\mathbf{u}_2\|^2$. Recalling that each $\Omega(i) \geq 0$ and each $\sigma_i(\cdot)$ is non-decreasing, and using (30) that $\mathbf{w}_1^T\mathbf{u}_1 \geq 0$, we have $\Delta(x) \geq 0$ for all $x \in \mathbb{R}$. Hence, (29) can be further lower-bounded by

$$|d_{\mathcal{F}_{nn}}(\mu_1,\nu_1) - d_{\mathcal{F}_{nn}}(\mu_2,\nu_2)| \geq \int_x \Delta(x)\varphi\left(x\,;\,0,\|\mathbf{w}_1\|^2\tau^2\right)\,\mathrm{d}x$$

$$\geq \int_0^{\frac{q(1)}{2}} \Delta(x)\varphi\left(x\,;\,0,\|\mathbf{w}_1\|^2\tau^2\right)\,\mathrm{d}x. \tag{31}$$

where $q(1)$ is defined in Assumption 2. Next, we develop a lower bound on the quantity $\Delta(x)$.

**Lemma 2.** *For $0 \leq x \leq q(1)/2$, we have*

$$\Delta(x) \geq \frac{M_F(1)M_F(d)\Gamma_{uB}}{\sqrt{3n}}\prod_{i=2}^{d-1}\Omega(i)\prod_{i=1}^{d-1}Q_\sigma(i),$$

*where $Q_\sigma(i), i = 1, 2, ..., d-1$ are defined in Assumption 2.*

*Proof.* Following from the definitions of $\mathbf{u}_1$ and $\mathbf{u}_2$ in (23), we have

$$\mathbf{w}_1^T\mathbf{u}_1 \overset{(i)}{\leq} \frac{M_F(1)\Gamma_{uB}}{\sqrt{3}}\sqrt{\frac{1}{n} + \frac{1}{m}} \overset{(ii)}{\leq} \frac{q(1)}{2} \tag{32}$$

where (i) follows from the inequality that $\|\mathbf{w}_1^T\mathbf{u}_1\| \leq \|\mathbf{w}_1\|\|\mathbf{u}_1\| = M_F(1)\|\mathbf{u}_1\|$ and (ii) follows from the assumption of Theorem 1 that $\sqrt{m^{-1}+n^{-1}} < \sqrt{3}q(1)/(2M_F(1)\Gamma_{\mathrm{uB}})$. Based on (32) that $\mathbf{w}_1^T\mathbf{u}_1 \leq q(1)/2$, we have, for any $0 \leq x \leq q(1)/2$,

$$0 \leq x + \mathbf{w}_1^T\mathbf{u}_1 \leq q(1),\ 0 \leq x < q(1)$$

which, using the definition of $\Omega(i)$ in (11) and letting $\Omega(d) = M_F(d)$, yields that, for $i = 2, 3, ..., d$

$$0 < \Omega(i)\sigma_{i-1}(\cdots\sigma_1(x)),\ \Omega(i)\sigma_{i-1}(\cdots\sigma_1(x+\mathbf{w}_1^T\mathbf{u}_1)) \leq \Omega(i)\sigma_{i-1}(\cdots\sigma_1(q(1))) \leq q(i).$$

Then, further by Assumption 2, we obtain

$$\Delta(x) \geq M_F(d)\left(\sigma_{d-1}(\cdots\sigma_1(x+\widetilde{\mathbf{w}}_1^T\mathbf{u}_1)) - \sigma_{d-1}(\cdots\sigma_1(x))\right)$$
$$\geq M_F(d)Q_\sigma(d-1)\Omega(d-1)\left(\sigma_{d-2}(\cdots\sigma_1(x+\widetilde{\mathbf{w}}_1^T\mathbf{u}_1)) - \sigma_{d-2}(\cdots\sigma_1(x))\right). \quad (33)$$

Repeating the step (33) for $d-1$ times, and using (30) that $\mathbf{w}_1^T\mathbf{u}_2 = 0$, we have

$$\Delta(x) \geq \mathbf{w}_1^T\mathbf{u}_1 M_F(d)\prod_{i=2}^{d-1}\Omega(i)\prod_{i=1}^{d-1}Q_\sigma(i) = \mathbf{w}_1^T(\mathbf{u}_1-\mathbf{u}_2)M_F(d)\prod_{i=2}^{d-1}\Omega(i)\prod_{i=1}^{d-1}Q_\sigma(i)$$

$$= M_F(1)M_F(d)\|\mathbf{u}_1-\mathbf{u}_2\|\prod_{i=2}^{d-1}\Omega(i)\prod_{i=1}^{d-1}Q_\sigma(i) = \frac{M_F(1)M_F(d)\Gamma_{\mathrm{uB}}}{\sqrt{3n}}\prod_{i=2}^{d-1}\Omega(i)\prod_{i=1}^{d-1}Q_\sigma(i),$$

which finishes the proof of Lemma 2. $\qquad\qquad\square$

Combining (31) and Lemma 2, we obtain

$$|d_{\mathcal{F}_{nn}}(\mu_1,\nu_1) - d_{\mathcal{F}_{nn}}(\mu_2,\nu_2)|$$

$$\geq \frac{M_F(1)M_F(d)\Gamma_{\mathrm{uB}}}{\sqrt{3n}}\prod_{i=2}^{d-1}\Omega(i)\prod_{i=1}^{d-1}Q_\sigma(i)\int_0^{\frac{q(1)}{2}}\varphi\left(x\,|\,0,\|\mathbf{w}_1\|^2\tau^2\right)\,\mathrm{d}x$$

$$= \frac{M_F(1)M_F(d)\Gamma_{\mathrm{uB}}}{\sqrt{3n}}\prod_{i=2}^{d-1}\Omega(i)\prod_{i=1}^{d-1}Q_\sigma(i)\int_0^{\frac{q(1)}{2\|\mathbf{w}_1\|\tau}}\varphi\left(x\,|\,0,1\right)\,\mathrm{d}x$$

$$\overset{(i)}{\geq} \frac{M_F(1)M_F(d)\Gamma_{\mathrm{uB}}}{\sqrt{3n}}\prod_{i=2}^{d-1}\Omega(i)\prod_{i=1}^{d-1}Q_\sigma(i)\int_0^{\frac{q(1)}{2M_F(1)\Gamma_{\mathrm{uB}}}}\varphi\left(x\,|\,0,1\right)\,\mathrm{d}x,$$

$$\geq \frac{M_F(1)M_F(d)\Gamma_{\mathrm{uB}}}{\sqrt{3n}}\prod_{i=2}^{d-1}\Omega(i)\prod_{i=1}^{d-1}Q_\sigma(i)\left(1-\Phi\left(\frac{q(1)}{2M_F(1)\Gamma_{\mathrm{uB}}}\right)\right), \quad (34)$$

where (i) follows from the fact that $\|\mathbf{w}_1\| = M_F(1)$, $\tau = \Gamma_{\mathrm{uB}}\sqrt{\frac{1}{3}\left(2+\frac{n}{m}\right)} \leq \Gamma_{\mathrm{uB}}$ and $\Phi(\cdot)$ is the CDF of the standard Gaussian distribution.

Next, we upper-bound the KL divergence between the distributions $\mathbb{P}_2$ and $\mathbb{P}_1$ as follows.

$$\mathrm{KL}(\mathbb{P}_2\|\mathbb{P}_1) = \mathrm{KL}(\mu_2^n\|\mu_1^n) + \mathrm{KL}(\nu_2^m\|\nu_1^m)$$
$$= n\mathrm{KL}(\mu_2\|\mu_1) + m\mathrm{KL}(\nu_2\|\nu_1)$$
$$= n\frac{\|\mathbf{u}_1\|^2}{2\tau^2} + m\frac{\|\mathbf{u}_2\|^2}{2\tau^2} = \frac{1}{2}. \quad (35)$$

Combining (34), (35) and Lemma 1 yields

$$\inf_{\hat{d}(n,m)}\sup_{\mu,\nu\in\mathcal{P}_{\mathrm{uB}}}\mathbb{P}\left\{\left|\hat{d}(n,m)-d_{\mathcal{F}_{nn}}(\mu,\nu)\right| \geq \frac{C(\mathcal{P}_{\mathrm{uB}})}{\sqrt{n}}\right\} \geq \max\{\frac{1}{4}e^{-1/2},\frac{1}{4}\} = \frac{1}{4}.$$

where $C(\mathcal{P}_{\mathrm{uB}})$ is the constant given by (10).

**Case 2: parameter set $\mathcal{W}_{1,\infty}$.** Recall from (5) that $\mathcal{W}_{1,\infty}$ is defined as

$$\mathcal{W}_{1,\infty} := \prod_{i=1}^{d-1}\left\{\mathbf{W}_i \in \mathbb{R}^{n_i\times m_i} : \|\mathbf{W}_i\|_{1,\infty} \leq M_{1,\infty}(i)\right\} \times \left\{\mathbf{w}_d \in \mathbb{R}^{n_d} : \|\mathbf{w}_d\|_1 \leq M_{1,\infty}(d)\right\}.$$

The proof for this case follows the steps similar to those in Case 1. To apply Lemma 1, we select four distributions $\mu_1, \nu_1, \mu_2$ and $\nu_2$ as in (22) with

$$\mathbf{u}_1 = [b_1, 0, 0, ..., 0]^T, \mathbf{u}_2 = [b_2, 0, 0, ..., 0]^T, \tau^2 = \frac{\Gamma_{\mathrm{uB}}^2}{3}\left(2 + \frac{n}{m}\right),$$

$$b_1^2 = \frac{\Gamma_{\mathrm{uB}}^2}{3}\left(\frac{1}{n} + \frac{1}{m}\right), b_2^2 = \frac{\Gamma_{\mathrm{uB}}^2}{3m}, (b_1 - b_2)^2 = \frac{\Gamma_{\mathrm{uB}}^2}{3n}. \tag{36}$$

Note that this construction also implies (23). Based on this construction, we pick the weights in $\hat{f}(\mathbf{x}) = \widehat{\mathbf{w}}_d^T \sigma_{d-1}\left(\cdots \sigma_1\left(\widehat{\mathbf{W}}_1 \mathbf{x}\right)\right) \in \mathcal{F}_{nn}$ as follows.

1. $\widehat{\mathbf{w}}_d(1) = M_{1,\infty}(d)$, $\widehat{\mathbf{w}}_d(s) = 0$ for $s \neq 1$,

2. For $i = 2, 3, ..., d-1, \widehat{\mathbf{W}}_i = [\widehat{\mathbf{w}}_i, \widehat{\mathbf{w}}_i, ..., \widehat{\mathbf{w}}_i]^T$, $\widehat{\mathbf{w}}_i(1) = M_{1,\infty}(i)$, $\widehat{\mathbf{w}}_i(s) = 0$ for $s \neq 1$,

3. $\widehat{\mathbf{W}}_1 = [\mathbf{w}_1, \mathbf{w}_1, ..., \mathbf{w}_1]^T$, $\mathbf{w}_1 = M_{1,\infty}(1)(\mathbf{u}_1 - \mathbf{u}_2)/\|\mathbf{u}_1 - \mathbf{u}_2\|$. $\tag{37}$

Clearly, (37) implies that $\|\widehat{\mathbf{w}}_d\|_1 = M_{1,\infty}(d)$, $\|\widehat{\mathbf{W}}_1\|_{1,\infty} = \|\widehat{\mathbf{w}}_1\|_1 = \|\widehat{\mathbf{w}}_1\| = M_{1,\infty}(1)$ and $\|\widehat{\mathbf{W}}_i\|_{1,\infty} = \|\widehat{\mathbf{w}}_i\|_1 = M_{1,\infty}(i)$ for $i = 2, ..., d-1$. Using the parameters chosen in (37), we have

$$\hat{f}(\mathbf{x}) = M_{1,\infty}(d)\sigma_{d-1}\left(\left(\widehat{\mathbf{w}}_{d-1}^T \mathbf{1}\right)\sigma_{d-2}\left(\cdots\left(\widehat{\mathbf{w}}_{d-1}^T \mathbf{1}\right)\sigma_1\left(\mathbf{w}_1^T \mathbf{x}\right)\right)\right)$$

$$= M_{1,\infty}(d)\sigma_{d-1}\left(M_{1,\infty}(d-1)\sigma_{d-2}\cdots M_{1,\infty}(2)\sigma_1\left(\mathbf{w}_1^T \mathbf{x}\right)\right), \tag{38}$$

where $\mathbf{1}$ denotes the all-one vector. Then, we have

$$|d_{\mathcal{F}_{nn}}(\mu_1, \nu_1) - d_{\mathcal{F}_{nn}}(\mu_2, \nu_2)| = d_{\mathcal{F}_{nn}}(\mu_1, \nu_1) = \sup_{f \in \mathcal{F}_{nn}} |\mathbb{E}_{\mathbf{x} \sim \mu_1} f(\mathbf{x}) - \mathbb{E}_{\mathbf{x} \sim \nu_1} f(\mathbf{x})|$$

$$\geq \left|\mathbb{E}_{\mathbf{x} \sim \mu_1} \hat{f}(\mathbf{x}) - \mathbb{E}_{\mathbf{x} \sim \nu_1} \hat{f}(\mathbf{x})\right|.$$

The remaining steps are the same as in Case 1, and are omitted.

## A.2 Proof of Corollary 1

Recall that if $\sigma_i(\cdot)$ is ReLU, then $q(i) \leq \infty$ and $Q_\sigma(i) = 1$. For the parameter set $\mathcal{W}_F$, we choose $q(1) = M_F(1)\Gamma_{\mathrm{uB}}$ and $q(i) = \infty$ for $i = 2, ..., d-1$ in Theorem 1. Then we obtain that $q(1)/(2M_F(1)\Gamma_{\mathrm{uB}}) = 0.5$ and $\Omega(i) = M_F(i)$, which, combined with $\sqrt{3}\left(1 - \Phi(0.5)\right)/6 > 0.08$, finish the proof. The result for the parameter set $\mathcal{W}_{1,\infty}$ can be proved in the same way.

# B Proof of Theorem 2

The proof is also based on the Le Cam's method (Lemma 1) as in Appendix A.1. However, here we deal with the bounded-support class of distributions. Hence, the hypothesis distributions we choose here are different. Suppose $m > n$ and the case $m < n$ follows the same steps.

**Case 1: parameter set is $\mathcal{W}_F$.** To use Lemma 1, we construct the following four distributions:

$$\mu_1(\mathbf{x}) = \begin{cases} \frac{1}{2} - \epsilon & \text{if } \mathbf{x} = \mathbf{x}_1 \\ \frac{1}{2} + \epsilon & \text{if } \mathbf{x} = -\mathbf{x}_1 \end{cases} \qquad \nu_1(\mathbf{x}) = \mu_2(\mathbf{x}) = \nu_2(\mathbf{x}) = \begin{cases} \frac{1}{2} & \text{if } \mathbf{x} = \mathbf{x}_1 \\ \frac{1}{2} & \text{if } \mathbf{x} = -\mathbf{x}_1 \end{cases} \tag{39}$$

where $\epsilon = \sqrt{2}n^{-\frac{1}{2}}/4 < 1/2$ and $\|\mathbf{x}_1\| = \Gamma_{\mathrm{B}}$.

First, we lower-bound $|d_{\mathcal{F}_{nn}}(\mu_1, \nu_1) - d_{\mathcal{F}_{nn}}(\mu_2, \nu_2)|$. Based on the construction (39), we obtain

$$|d_{\mathcal{F}_{nn}}(\mu_1, \nu_1) - d_{\mathcal{F}_{nn}}(\mu_2, \nu_2)| = d_{\mathcal{F}_{nn}}(\mu_1, \nu_1)$$

$$= \sup_{f \in \mathcal{F}_{nn}} |\mathbb{E}_{\mathbf{x} \sim \mu_1} f(\mathbf{x}) - \mathbb{E}_{\mathbf{x} \sim \nu_1} f(\mathbf{x})|$$

$$= \sup_{f \in \mathcal{F}_{nn}} \left|\left(\frac{1}{2} - \epsilon\right) f(\mathbf{x}_1) + \left(\frac{1}{2} + \epsilon\right) f(-\mathbf{x}_1) - \frac{1}{2}f(\mathbf{x}_1) - \frac{1}{2}f(-\mathbf{x}_1)\right|$$

$$= \epsilon \sup_{f \in \mathcal{F}_{nn}} |f(\mathbf{x}_1) - f(-\mathbf{x}_1)|$$

$$\geq \epsilon |\tilde{f}(\mathbf{x}_1) - \tilde{f}(-\mathbf{x}_1)|$$

$$= \epsilon\left(M_F(d)\sigma_{d-1}(\cdots\sigma_1(M_F(1)\Gamma_{\mathrm{B}})) - M_F(d)\sigma_{d-1}(\cdots\sigma_1(-M_F(1)\Gamma_{\mathrm{B}}))\right), \tag{40}$$

where the function $\tilde{f}(\mathbf{x}) \in \mathcal{F}_{nn}$ is constructed using an approach similar to (26), which is given by

$$\tilde{f}(\mathbf{x}) = M_F(d)\sigma_{d-1}\left(M_F(d-1)\cdots M_F(2)\sigma_1\left(\mathbf{w}_1^T\mathbf{x}\right)\right) \text{ with } \mathbf{w}_1 = M_F(1)\mathbf{x}_1/\|\mathbf{x}_1\|.$$

Next, we derive an upper bound on the KL divergence between the distributions as follows.

$$
\begin{aligned}
\mathrm{KL}(\mathbb{P}_2\|\mathbb{P}_1) &= n\mathrm{KL}(\mu_2\|\mu_1) + m\mathrm{KL}(\nu_2\|\nu_1) \\
&\overset{(i)}{=} n\left(\frac{1}{2}\log\left(\frac{1}{1-2\epsilon}\right) + \frac{1}{2}\log\left(\frac{1}{1+2\epsilon}\right)\right) \\
&= \frac{1}{2}n\log\left(1 + \frac{4\epsilon^2}{1-4\epsilon^2}\right) \overset{(ii)}{\leq} \frac{1}{2}n\frac{4\epsilon^2}{1-4\epsilon^2} \\
&\overset{(iii)}{\leq} \frac{1}{4-2/n} \leq \frac{1}{2},
\end{aligned}
\tag{41}
$$

where (i) follows from the fact that $\nu_1 = \nu_2$, (ii) follows from the inequality that $\log(1 + x) \leq x$ for $x > 0$ and (iii) follows because $\epsilon = \sqrt{2}n^{-\frac{1}{2}}/4$. Hence, combining (40), (41), $\epsilon = \sqrt{2}n^{-\frac{1}{2}}/4$ and Lemma 1, and noting that $\sqrt{2}/8 > 0.17$, we complete the proof.

**Case 2: parameter set is $\mathcal{W}_{1,\infty}$.** Similarly to the proof for Case 1, we select the same distributions as in (39) with the parameters satisfying

$$\mathbf{x}_1 = [\Gamma_{\mathbf{B}}, 0, 0, ..., 0]^T, \quad \epsilon = \sqrt{2}n^{-1/2}/4. \tag{42}$$

Clearly, (42) implies $\|\mathbf{x}_1\|_1 = \|\mathbf{x}_1\| = \Gamma_{\mathbf{B}}$. Using an approach similar to (40), we obtain

$$
\begin{aligned}
|d_{\mathcal{F}_{nn}}(\mu_1, \nu_1) - d_{\mathcal{F}_{nn}}(\mu_2, \nu_2)| &= d_{\mathcal{F}_{nn}}(\mu_1, \nu_1) \\
&= \sup_{f \in \mathcal{F}_{nn}} |\mathbb{E}_{\mathbf{x}\sim\mu_1}f(\mathbf{x}) - \mathbb{E}_{\mathbf{x}\sim\nu_1}f(\mathbf{x})| \\
&= \epsilon \sup_{f \in \mathcal{F}_{nn}} |f(\mathbf{x}_1) - f(-\mathbf{x}_1)| \\
&\geq \epsilon |\hat{f}(\mathbf{x}_1) - \hat{f}(-\mathbf{x}_1)| \\
&= \epsilon\left(M_{1,\infty}(d)\sigma_{d-1}(\cdots\sigma_1(M_{1,\infty}(1)\Gamma_{\mathbf{B}})) - M_{1,\infty}(d)\sigma_{d-1}(\cdots\sigma_1(-M_{1,\infty}(1)\Gamma_{\mathbf{B}}))\right), \quad (43)
\end{aligned}
$$

where the function $\hat{f}(\mathbf{x}) \in \mathcal{F}_{nn}$ is constructed based on a approach similar to (38), which is given by

$$\hat{f}(\mathbf{x}) = M_{1,\infty}(d)\sigma_{d-1}\left(\cdots M_{1,\infty}(2)\sigma_1\left(\mathbf{w}_1^T\mathbf{x}\right)\right) \text{ with } \mathbf{w}_1 = M_{1,\infty}(1)\mathbf{x}_1/\|\mathbf{x}_1\|. \tag{44}$$

Substituting $\epsilon = \sqrt{2}n^{-1/2}/4$ into (43) and adopting the same steps in (41), we finish the proof by Lemma 1.

## C  Proof of Theorem 3

As we outline in Section 3.4, the proof of Theorem 3 follows from the proofs of Theorems 5, 6 and 7 as three main steps. We next provide the proofs for these theorems in three subsections.

### C.1  Proof of Theorem 5

The proof follows from the general idea in [11] for the scalar case. The major technical development here lies in upper-bounding $\mathbb{E}\left(e^{\lambda\mathbf{V}_i}\big|\mathbf{x}_1, ..., \mathbf{x}_{i-1}\right)$ for the martingale difference $\mathbf{V}_i$ based on a tail bound of sub-Gaussian random vectors, and then using the bound of $\mathbb{E}\left(e^{\lambda\mathbf{V}_i}\big|\mathbf{x}_1, ..., \mathbf{x}_{i-1}\right)$ to yield (15) by Markov's inequality. To simplify the notation in the proof, we use $\mathbf{x}_{n+1}, ..., \mathbf{x}_{n+m}$ to denote $\mathbf{y}_1, ..., \mathbf{y}_m$.

Let $\mathbf{V}_i = \mathbb{E}_{\mathbf{x}_1,\ldots,\mathbf{x}_{n+m}}(F|\mathbf{x}_1,\ldots,\mathbf{x}_i) - \mathbb{E}_{\mathbf{x}_1,\ldots,\mathbf{x}_{n+m}}(F|\mathbf{x}_1,\ldots,\mathbf{x}_{i-1})$. Then, we have

$$
\begin{aligned}
\mathbb{E}\left(e^{\lambda \mathbf{V}_i}\big|\mathbf{x}_1,\ldots,\mathbf{x}_{i-1}\right) &= \int_{\mathbf{x}_i} e^{\lambda \mathbb{E}(F|\mathbf{x}_1,\ldots,\mathbf{x}_i) - \lambda \mathbb{E}(F|\mathbf{x}_1,\ldots,\mathbf{x}_{i-1})} \mathrm{d}\,\mathbb{P}_{\mathbf{x}_i} \\
&\overset{(i)}{\leq} \int_{\mathbf{x}_i} \mathbb{E}_{\mathbf{x}_{i+1},\ldots,\mathbf{x}_{n+m}} e^{\lambda F} \, \mathbb{E}_{\mathbf{x}_i,\ldots,\mathbf{x}_{n+m}} e^{-\lambda F} \mathrm{d}\,\mathbb{P}_{\mathbf{x}_i} \\
&= \int_{\mathbf{x}_i} \left( \int_{\mathbf{x}_{i+1},\ldots,\mathbf{x}_{n+m}} e^{\lambda F} \mathrm{d}\,\mathbb{P}_{\mathbf{x}_{i+1}} \cdots \mathrm{d}\,\mathbb{P}_{\mathbf{x}_{n+m}} \int_{\mathbf{x}_i',\ldots,\mathbf{x}_{n+m}} e^{-\lambda F} \mathrm{d}\,\mathbb{P}_{\mathbf{x}_i'} \cdots \mathrm{d}\,\mathbb{P}_{\mathbf{x}_{n+m}} \right) \mathrm{d}\,\mathbb{P}_{\mathbf{x}_i} \\
&= \int_{\mathbf{x}_{i+1},\ldots,\mathbf{x}_{n+m}} \left( \int_{\mathbf{x}_i,\mathbf{x}_i'} e^{\lambda F(\ldots,\mathbf{x}_i,\ldots) - \lambda F(\ldots,\mathbf{x}_i',\ldots)} \mathrm{d}\,\mathbb{P}_{\mathbf{x}_i} \mathrm{d}\,\mathbb{P}_{\mathbf{x}_i'} \right) \mathrm{d}\,\mathbb{P}_{\mathbf{x}_{i+1}} \cdots \mathrm{d}\,\mathbb{P}_{\mathbf{x}_{n+m}}, \quad (45)
\end{aligned}
$$

where (i) follows from the Jensen's inequality. Then, using the fact that $e^t + e^{-t} \leq e^{-s} + e^s$ for $\forall |t| \leq s$ and noting (14), we have, for $1 \leq i \leq n$,

$$
e^{\lambda(F(\ldots,\mathbf{x}_i,\ldots) - F(\ldots,\mathbf{x}_i',\ldots))} + e^{-\lambda(F(\ldots,\mathbf{x}_i,\ldots) - F(\ldots,\mathbf{x}_i',\ldots))} \leq e^{\frac{\lambda L_{\mathcal{F}}\|\mathbf{x}_i - \mathbf{x}_i'\|}{n}} + e^{-\frac{\lambda L_{\mathcal{F}}\|\mathbf{x}_i - \mathbf{x}_i'\|}{n}}. \quad (46)
$$

Thus, we have, for $1 \leq i \leq n$,

$$
\begin{aligned}
&\int_{\mathbf{x}_i,\mathbf{x}_i'} e^{\lambda F(\mathbf{x}_1,\ldots,\mathbf{x}_i,\ldots,\mathbf{x}_{n+m}) - \lambda F(\mathbf{x}_1,\ldots,\mathbf{x}_i',\ldots,\mathbf{x}_{n+m})} \mathrm{d}\,\mathbb{P}_{\mathbf{x}_i} \mathrm{d}\,\mathbb{P}_{\mathbf{x}_i'} \\
&\overset{(i)}{=} \frac{1}{2} \int_{\mathbf{x}_i,\mathbf{x}_i'} e^{\lambda(F(\ldots,\mathbf{x}_i,\ldots) - F(\ldots,\mathbf{x}_i',\ldots))} + e^{-\lambda(F(\ldots,\mathbf{x}_i,\ldots) - F(\ldots,\mathbf{x}_i',\ldots))} \mathrm{d}\,\mathbb{P}_{\mathbf{x}_i} \mathrm{d}\,\mathbb{P}_{\mathbf{x}_i'} \\
&\overset{(ii)}{\leq} \frac{1}{2} \int_{\mathbf{x}_i,\mathbf{x}_i'} e^{\lambda L_{\mathcal{F}}\|\mathbf{x}_i - \mathbf{x}_i'\|/n} + e^{-\lambda L_{\mathcal{F}}\|\mathbf{x}_i - \mathbf{x}_i'\|/n} \mathrm{d}\,\mathbb{P}_{\mathbf{x}_i} \mathrm{d}\,\mathbb{P}_{\mathbf{x}_i'} \\
&= \frac{1}{2}\mathbb{E}_{\mathbf{x}_i,\mathbf{x}_i'} \left( e^{\lambda L_{\mathcal{F}}\|\mathbf{x}_i - \mathbf{x}_i'\|/n} + e^{-\lambda L_{\mathcal{F}}\|\mathbf{x}_i - \mathbf{x}_i'\|/n} \right) \\
&\overset{(iii)}{\leq} \mathbb{E}_{\mathbf{x}_i,\mathbf{x}_i'} e^{\lambda^2 L_{\mathcal{F}}^2 \|\mathbf{x}_i - \mathbf{x}_i'\|^2/(2n^2)} = \mathbb{E}_{\mathbf{z}_i} e^{\lambda^2 L_{\mathcal{F}}^2 \|\mathbf{z}_i\|^2/(2n^2)}
\end{aligned} \quad (47)
$$

where (i) follows from the symmetry between $\mathbf{x}_i$ and $\mathbf{x}_i'$, (ii) is based on (46), (iii) follows from the inequality that $(e^x + e^{-x})/2 \leq e^{x^2/2}$, and we set $\mathbf{z}_i = \mathbf{x}_i - \mathbf{x}_i'$ in the last equality. Since $\mathbf{x}_i$ and $\mathbf{x}_i$ are both sub-Gaussian with mean $\mathbf{u}_x$ and variance parameter $\tau_x$, we have $\mathbb{E}(\mathbf{z}_i) = 0$ and

$$
\mathbb{E}\, e^{\mathbf{a}^T \mathbf{z}_i} = \mathbb{E}\, e^{\mathbf{a}^T \mathbf{x}_i}\, \mathbb{E}\, e^{-\mathbf{a}^T \mathbf{x}_i'} \leq e^{\|\mathbf{a}\|^2 \sigma^2/2} e^{\|\mathbf{a}\|^2 \sigma_x^2/2} = e^{\|\mathbf{a}\|^2 \sigma_x^2}, \quad (48)
$$

which implies that $\mathbf{z}_i$ is a zero-mean sub-Gaussian random variable with the variance parameter $2\sigma_x^2$. Next, we quote the following tail inequality from [10], which is useful here.

**Lemma 3** (Theorem 1 in [10]). *Let $\mathbf{A} \in \mathbb{R}^{h \times h}$ be a matrix and let $\mathbf{\Sigma} = \mathbf{A}^T \mathbf{A}$. Suppose that $\mathbf{x}$ is a sub-Gaussian random vector with mean $\mathbf{u} \in \mathbb{R}^h$ and variance parameter $\tau^2$. Then, for $0 \leq \eta < 1/(2\tau^2 \|\mathbf{\Sigma}\|)$,*

$$
\mathbb{E}\left(e^{\eta \|\mathbf{A}\mathbf{x}\|^2}\right) \leq \exp\left(\tau^2 \operatorname{tr}(\mathbf{\Sigma})\eta + \frac{\tau^4 \operatorname{tr}(\mathbf{\Sigma}^2)\eta^2 + \|\mathbf{A}\mathbf{u}\|^2 \eta}{1 - 2\tau^2 \|\mathbf{\Sigma}\|\eta}\right). \quad (49)
$$

Recall from (48) that $\mathbf{z}_i$ is sub-Gaussian with variance parameter $2\tau_x^2$ and mean $\mathbf{0}$. Then, letting $\mathbf{A} = \mathbf{I}_h$ and $\mathbf{u} = \mathbf{0}$ in Lemma 3 and using (47), we have, for $n \geq \sqrt{2}\tau_x \lambda L_{\mathcal{F}}$,

$$
\mathbb{E}\, \exp\left(\frac{\lambda^2 L_{\mathcal{F}}^2}{2n^2}\|\mathbf{z}_i\|^2\right) \leq \exp\left(\frac{\tau_x^2 h \lambda^2 L_{\mathcal{F}}^2}{n^2} + \frac{\tau_x^4 h \lambda^4 L_{\mathcal{F}}^4}{n^4 - 2\tau_x^2 \lambda^2 L_{\mathcal{F}}^2 n^2}\right). \quad (50)
$$

Assume that $n \geq \sqrt{3}\tau_x \lambda L_{\mathcal{F}}$. Then, (50) can be further upper-bounded by

$$
\mathbb{E}\, \exp\left(\frac{\lambda^2 L_{\mathcal{F}}^2}{2n^2}\|\mathbf{z}_i\|^2\right) \leq \exp\left(\frac{2\tau_x^2 h \lambda^2 L_{\mathcal{F}}^2}{n^2}\right),
$$

which, in conjunction with (45) and (47), implies that for $1 \leq i \leq n$,

$$
\mathbb{E}\left(e^{\lambda \mathbf{V}_i}\big|\mathbf{x}_1,\ldots,\mathbf{x}_{i-1}\right) \leq \exp\left(2\tau_x^2 h \lambda^2 L_{\mathcal{F}}^2/n^2\right). \quad (51)
$$

Using similar steps, we obtain, for $n + 1 \leq i \leq n + m$ and $m \geq \sqrt{3}\tau_y \lambda L_{\mathcal{F}}$,

$$\mathbb{E}\left(e^{\lambda \mathbf{V}_i} \big| \mathbf{x}_1, ..., \mathbf{x}_{i-1}\right) \leq \exp\left(2\tau_y^2 h\lambda^2 L_{\mathcal{F}}^2/m^2\right). \tag{52}$$

Then, using (51), (52) and Markov's inequality, we have

$$\mathbb{P}\big(F(\mathbf{x}_1, ..., \mathbf{x}_{n+m}) - \mathbb{E}\, F(\mathbf{x}_1, ..., \mathbf{x}_{n+m}) \geq \epsilon\big)$$

$$= \mathbb{P}\left(\sum_{i=1}^{n+m} \mathbf{V}_i > \epsilon\right) \leq e^{-\lambda\epsilon} \mathbb{E}_{\mathbf{x}_1, ..., \mathbf{x}_{n+m}} \left(\prod_{i=1}^{n+m} e^{\lambda \mathbf{V}_i}\right)$$

$$= e^{-\lambda\epsilon} \mathbb{E}_{\mathbf{x}_1, ..., \mathbf{x}_{n+m-1}} \left(\mathbb{E}_{\mathbf{x}_{n+m}} \left(\prod_{i=1}^{n+m} e^{\lambda \mathbf{V}_i} | \mathbf{x}_1, \cdots, \mathbf{x}_{n+m-1}\right)\right)$$

$$= e^{-\lambda\epsilon} \mathbb{E}_{\mathbf{x}_1, ..., \mathbf{x}_{n+m-1}} \left(\prod_{i=1}^{n+m-1} e^{\lambda \mathbf{V}_i} \mathbb{E}_{\mathbf{x}_{n+m}} \left(e^{\lambda \mathbf{V}_{n+m}} | \mathbf{x}_1, \cdots, \mathbf{x}_{n+m-1}\right)\right)$$

$$\overset{(i)}{\leq} e^{-\lambda\epsilon} e^{2\tau_y^2 h\lambda^2 L_{\mathcal{F}}^2/m^2} \mathbb{E}_{\mathbf{x}_1, ..., \mathbf{x}_{n+m-1}} \left(\prod_{i=1}^{n+m-1} e^{\lambda \mathbf{V}_i}\right)$$

$$\overset{(ii)}{\leq} \exp\left(-\lambda\epsilon + 2\tau_y^2 h\lambda^2 L_{\mathcal{F}}^2/m + 2\tau_x^2 h\lambda^2 L_{\mathcal{F}}^2/n\right)$$

$$\overset{(iii)}{\leq} \exp\left(-\lambda\epsilon + 2\Gamma_{\text{uB}}^2 h\lambda^2 L_{\mathcal{F}}^2/m + 2\Gamma_{\text{uB}}^2 h\lambda^2 L_{\mathcal{F}}^2/n\right), \tag{53}$$

where (ii) follows by repeating (i) for $n + m - 1$ times and (iii) follows from the fact that $0 < \tau_x, \tau_y \leq \Gamma_{\text{uB}}$. Let $M = \Gamma_{\text{uB}}^2 h L_{\mathcal{F}}^2$. Optimizing (53) over $\lambda$, we have $\lambda = \epsilon M^{-1}(1/n + 1/m)^{-1}$ and

$$\mathbb{P}\left(F(\mathbf{x}_1, ..., \mathbf{x}_{n+m}) - \mathbb{E}\, F(\mathbf{x}_1, ..., \mathbf{x}_{n+m}) \geq \epsilon\right) \leq \exp\left(\frac{-\epsilon^2 nm}{8M(n+m)}\right). \tag{54}$$

Recall that our proof requires that $n \geq \sqrt{3}\tau_x \lambda L_{\mathcal{F}}$ and $m \geq \sqrt{3}\tau_y \lambda L_{\mathcal{F}}$, which, based on the facts that $\lambda = \epsilon M^{-1}(1/n + 1/m)^{-1}$ and $0 < \tau_x, \tau_y \leq \Gamma_{\text{uB}}$, are satisfied for any $0 < \epsilon \leq \sqrt{3}h\Gamma_{\text{uB}}L_{\mathcal{F}} \min\{m, n\}(n^{-1} + m^{-1})$. Thus, the proof is complete.

### C.2 Proof of Theorem 6

Based on the definition of $d_{\mathcal{F}_{nn}}(\mu, \nu)$, we have

$$|d_{\mathcal{F}_{nn}}(\mu, \nu) - d_{\mathcal{F}_{nn}}(\hat{\mu}, \hat{\nu})| = \left| \sup_{f \in \mathcal{F}_{nn}} |\mathbb{E}_{\mathbf{x}\sim\mu}f(\mathbf{x}) - \mathbb{E}_{\mathbf{y}\sim\nu}f(\mathbf{y})| - \sup_{f \in \mathcal{F}_{nn}} |\mathbb{E}_{\mathbf{x}\sim\hat{\mu}}f(\mathbf{x}) - \mathbb{E}_{\mathbf{y}\sim\hat{\nu}}f(\mathbf{y})| \right|$$

$$\leq \underbrace{\sup_{f \in \mathcal{F}_{nn}} \left| \mathbb{E}_{\mathbf{x}\sim\mu}f(\mathbf{x}) - \frac{1}{n}\sum_{i=1}^{n} f(\mathbf{x}_i) - \left(\mathbb{E}_{\mathbf{y}\sim\nu}f(\mathbf{y}) - \frac{1}{m}\sum_{i=1}^{m} f(\mathbf{y}_i)\right) \right|}_{F(\mathbf{x}_1, ..., \mathbf{x}_n, \mathbf{y}_1, ..., \mathbf{y}_m)}. \tag{55}$$

First note that for $\forall\, 1 \leq i \leq n$,

$$|F(\mathbf{x}_1, ..., \mathbf{x}_i, ..., \mathbf{y}_m) - F(\mathbf{x}_1, ..., \mathbf{x}_i', ..., \mathbf{y}_m)| \leq \sup_{f \in \mathcal{F}_{nn}} |f(\mathbf{x}_i) - f(\mathbf{x}_i')| / n. \tag{56}$$

If the parameter set is $\mathcal{W}_F$, then using Cauchy-Schwarz inequality, we have

$$|f(\mathbf{x}_i) - f(\mathbf{x}_i')| = |\mathbf{w}_d^T \sigma_{d-1}(\cdots \sigma_1(\mathbf{W}_1 \mathbf{x}_i)) - \mathbf{w}_d^T \sigma_{d-1}(\cdots \sigma_1(\mathbf{W}_1 \mathbf{x}_i'))|$$

$$\leq M_F(d)\|\sigma_{d-1}(\cdots \sigma_1(\mathbf{W}_1 \mathbf{x}_i)) - \sigma_{d-1}(\cdots \sigma_1(\mathbf{W}_1 \mathbf{x}_i'))\|$$

$$\overset{(i)}{\leq} M_F(d)L_{d-1}\|\mathbf{W}_{d-1}\sigma_{d-2}(\cdots \sigma_1(\mathbf{W}_1 \mathbf{x}_i)) - \mathbf{W}_{d-1}\sigma_{d-2}(\cdots \sigma_1(\mathbf{W}_1 \mathbf{x}_i))\|$$

$$\leq M_F(d)L_{d-1}M_F(d-1)\|\sigma_{d-2}(\cdots \sigma_1(\mathbf{W}_1 \mathbf{x}_i)) - \sigma_{d-2}(\cdots \sigma_1(\mathbf{W}_1 \mathbf{x}_i'))\|, \tag{57}$$

where (i) follows from the fact that $\sigma_{d-1}(\cdot)$ is $L_{d-1}$-Lipschitz. Repeating the process (57), we obtain

$$|f(\mathbf{x}_i) - f(\mathbf{x}_i')| \leq \prod_{i=1}^{d} M_F(i) \prod_{i=1}^{d-1} L_i \|\mathbf{x}_i - \mathbf{x}_i'\|,$$

which, in conjunction with (56), implies that, for $\forall\, 1 \leq i \leq n$

$$|F(\mathbf{x}_1, ..., \mathbf{x}_i, ..., \mathbf{y}_m) - F(\mathbf{x}_1, ..., \mathbf{x}_i', ..., \mathbf{y}_m)| \leq \prod_{i=1}^{d} M_F(i) \prod_{i=1}^{d-1} L_i \|\mathbf{x}_i - \mathbf{x}_i'\|/n. \qquad (58)$$

Similarly, we can get, for $\forall\, 1 \leq i \leq m$,

$$|F(\mathbf{x}_1, ..., \mathbf{y}_i, ..., \mathbf{y}_m) - F(\mathbf{x}_1, ..., \mathbf{y}_i', ..., \mathbf{y}_m)| \leq \prod_{i=1}^{d} M_F(i) \prod_{i=1}^{d-1} L_i \|\mathbf{y}_i - \mathbf{y}_i'\|/m. \qquad (59)$$

Let $K := \prod_{i=1}^{d} M_F(i) \prod_{i=1}^{d-1} L_i$. Combining (58), (59) and Theorem 5, we have, for any $0 < \epsilon \leq \sqrt{3}h\Gamma_{\mathrm{uB}}K\min\{m,n\}(n^{-1}+m^{-1})$,

$$\mathbb{P}\left(F(\mathbf{x}_1, ..., \mathbf{x}_n, ...., \mathbf{y}_m) - \mathbb{E}\,F(\mathbf{x}_1, ..., \mathbf{x}_n, ...., \mathbf{y}_m) \geq \epsilon\right) \leq \exp\left(\frac{-\epsilon^2 mn}{8h\Gamma_{\mathrm{uB}}^2 K^2(m+n)}\right), \quad (60)$$

where $\Gamma_{\mathrm{uB}}$ is defined in (8). Plugging $\delta = \exp\left(-\epsilon^2 mn/(8h\Gamma_{\mathrm{uB}}^2 K^2(m+n))\right)$ in (60) implies that, if $\sqrt{6h}\min\{n,m\}\sqrt{m^{-1}+n^{-1}} \geq 4\sqrt{\log(1/\delta)}$, then with probability at least $1-\delta$,

$$F(\mathbf{x}_1, ...., \mathbf{y}_m) \leq \mathbb{E}\,F(\mathbf{x}_1, ...., \mathbf{y}_m) + 2\Gamma_{\mathrm{uB}} \prod_{i=1}^{d} M_F(i) \prod_{i=1}^{d-1} L_i \sqrt{2h\left(\frac{1}{n}+\frac{1}{m}\right)\log\frac{1}{\delta}}. \qquad (61)$$

Next, we upper-bound the expectation term in (61) through the following steps.

$$\mathbb{E}\,F(\mathbf{x}_1, ..., \mathbf{x}_n, ...., \mathbf{y}_m)$$

$$=\mathbb{E}_{\{\mathbf{x}_i\},\{\mathbf{y}_i\}} \sup_{f \in \mathcal{F}_{nn}} \left|\mathbb{E}_{\mathbf{x}\sim\mu}f(\mathbf{x}) - \frac{1}{n}\sum_{i=1}^{n} f(\mathbf{x}_i) - \left(\mathbb{E}_{\mathbf{y}\sim\nu}f(\mathbf{y}) - \frac{1}{m}\sum_{i=1}^{m} f(\mathbf{y}_i)\right)\right|$$

$$\leq \mathbb{E}_{\mathbf{x},\mathbf{y},\mathbf{x}',\mathbf{y}',\epsilon,\epsilon'} \sup_{f \in \mathcal{F}_{nn}} \left|\frac{1}{n}\sum_{i=1}^{n}\epsilon_i\left(f(\mathbf{x}_i') - f(\mathbf{x}_i)\right) - \frac{1}{m}\sum_{i=1}^{m}\epsilon_i'\left(f(\mathbf{y}_i') - f(\mathbf{y}_i)\right)\right|$$

$$\leq \mathbb{E}_{\mathbf{x},\mathbf{x}',\epsilon} \sup_{f \in \mathcal{F}_{nn}} \left|\frac{1}{n}\sum_{i=1}^{n}\epsilon_i\left(f(\mathbf{x}_i') - f(\mathbf{x}_i)\right)\right| + \mathbb{E}_{\mathbf{y},\mathbf{y}',\epsilon'} \sup_{f \in \mathcal{F}_{nn}} \left|\frac{1}{m}\sum_{i=1}^{m}\epsilon_i\left(f(\mathbf{y}_i') - f(\mathbf{y}_i)\right)\right|$$

$$\leq 2\mathcal{R}_n(\mathcal{F}_{nn}, \mu) + 2\mathcal{R}_m(\mathcal{F}_{nn}, \nu) \qquad (62)$$

which, combined with (55) and (61), finishes the proof for the parameter set $\mathcal{W}_F$.

If the parameter set is $\mathcal{W}_{1,\infty}$, then we have

$$|f(\mathbf{x}_i) - f(\mathbf{x}_i')| = |\mathbf{w}_d^T \sigma_{d-1}\left(\cdots\sigma_1(\mathbf{W}_1\mathbf{x}_i)\right) - \mathbf{w}_d^T \sigma_{d-1}\left(\cdots\sigma_1(\mathbf{W}_1\mathbf{x}_i')\right)|$$

$$\overset{(i)}{\leq} \|\mathbf{w}_d\|_1 \|\sigma_{d-1}\left(\cdots\sigma_1(\mathbf{W}_1\mathbf{x}_i)\right) - \sigma_{d-1}\left(\cdots\sigma_1(\mathbf{W}_1\mathbf{x}_i')\right)\|_\infty$$

$$\leq M_{1,\infty}(d)L_{d-1}\|\mathbf{W}_{d-1}\sigma_{d-2}\left(\cdots\sigma_1(\mathbf{W}_1\mathbf{x}_i)\right) - \mathbf{W}_{d-1}\sigma_{d-2}\left(\cdots\sigma_1(\mathbf{W}_1\mathbf{x}_i)\right)\|_\infty$$

$$\overset{(ii)}{\leq} M_{1,\infty}(d)L_{d-1}M_{1,\infty}(d-1)\|\sigma_{d-2}\left(\cdots\sigma_1(\mathbf{W}_1\mathbf{x}_i)\right) - \sigma_{d-2}\left(\cdots\sigma_1(\mathbf{W}_1\mathbf{x}_i)\right)\|_\infty$$

$$\leq \prod_{i=1}^{d} M_{1,\infty}(i) \prod_{i=1}^{d-1} L_i \|\mathbf{x}_i - \mathbf{x}_i'\|_\infty \leq \prod_{i=1}^{d} M_{1,\infty}(i) \prod_{i=1}^{d-1} L_i \|\mathbf{x}_i - \mathbf{x}_i'\|, \qquad (63)$$

where (i) follows from the inequality that $\mathbf{w}^T\mathbf{x} \leq \|\mathbf{w}\|_1\|\mathbf{x}\|_\infty$ and (ii) follows from $\mathbf{W}\mathbf{x} \leq \|\mathbf{W}\|_{1,\infty}\|\mathbf{x}\|_\infty$. The remaining steps are the same as in the case when the parameter set is $\mathcal{W}_F$, and are omitted.

### C.3 Proof of Theorem 7

As commented in Section 3.4, directly applying the existing results on the Rademacher complexity of neural networks in [8] to unbounded sub-Gaussian inputs can lead to a loose upper bound. Hence,

here we use a different approach that takes advantage of the sub-Gaussianity of the input data. Our technique first upper-bounds the Rademacher complexity $\mathbb{E}_{\epsilon,\mathbf{x}} \sup_{f\in\mathcal{F}_{nn}} |\sum_{i=1}^n \epsilon_i f(\mathbf{x}_i)/n|$ by

$$\sqrt{\lambda \log\left(\mathbb{E}_{\epsilon,\mathbf{x}} \sup_{f\in\mathcal{F}_{nn}} \exp\left(\lambda \left|\sum_{i=1}^n \epsilon_i f(\mathbf{x}_i)/n\right|^2\right)\right)}, \tag{64}$$

and then upper-bounds the expectation term in (64) by combining a peeling method different from that in [8] and a tail bound of sub-Gaussian random vectors.

To prove Theorem 7, we first establish a useful lemma as follows.

**Lemma 4.** *For any input $\mathbf{z} \in \mathbb{R}^t$, let $\sigma(\mathbf{z}) := [\sigma(z_1), \sigma(z_2), ..., \sigma(z_t)]^T$. Then, we have*

**(I)** *If the acitvatio function $\sigma(\cdot)$ is L-Lipchitz and satisfies $\sigma(\alpha x) = \alpha\sigma(x)$ for all $\alpha \geq 0$, then for any vector-valued function class $\mathcal{F}$ and any constant $\eta > 0$,*

$$\mathbb{E}_{\mathbf{x},\epsilon} \sup_{f\in\mathcal{F}, \|\mathbf{W}\|_F\leq R} \exp\left(\eta\left\|\sum_{i=1}^n \epsilon_i\sigma(\mathbf{W}f(\mathbf{x}_i))\right\|^2\right) \leq 2\mathbb{E}_{\mathbf{x},\epsilon} \sup_{f\in\mathcal{F}} \exp\left(\eta R^2 L^2 \left\|\sum_{i=1}^n \epsilon_i f(\mathbf{x}_i)\right\|^2\right).$$

**(II)** *If the acitvatio function $\sigma(\cdot)$ is L-Lipchitz and satisfies $\sigma(0) = 0$, then for any vector-valued function class $\mathcal{F}$ and any constant $\eta > 0$,*

$$\mathbb{E}_{\mathbf{x},\epsilon} \sup_{f\in\mathcal{F}, \|\mathbf{W}\|_{1,\infty}\leq R} \exp\left(\eta\left\|\sum_{i=1}^n \epsilon_i\sigma(\mathbf{W}f(\mathbf{x}_i))\right\|_\infty\right) \leq 2\mathbb{E}_{\mathbf{x},\epsilon} \sup_{f\in\mathcal{F}} \exp\left(\eta R^2 L^2 \left\|\sum_{i=1}^n \epsilon_i f(\mathbf{x}_i)\right\|_\infty\right).$$

*Proof.* The proof of the first result follows the general idea of that of Lemma 1 in [8]. However, we cannot directly apply Lemma 1 in [8] because the function $\exp(\eta x^2)$ is not increasing over entire $\mathbb{R}$. Thus, we need to tailor its proof to our setting.

Consider a function $g : \mathbb{R} \longmapsto (0,\infty)$ given by $g(x) = \exp(\eta x^2)\mathbf{I}(x \geq 0) + \mathbf{I}(x < 0)$, where $\mathbf{I}(\cdot)$ is the indicator function. It can be verified that $g(\cdot)$ is increasing and convex. Then, we have

$$\mathbb{E}_{\mathbf{x},\epsilon} \sup_{f\in\mathcal{F}, \|\mathbf{W}\|_F\leq R} \exp\left(\eta\left\|\sum_{i=1}^n \epsilon_i\sigma(\mathbf{W}f(\mathbf{x}_i))\right\|^2\right) = \mathbb{E}_{\mathbf{x},\epsilon} \sup_{f\in\mathcal{F}, \|\mathbf{W}\|_F\leq R} g\left(\left\|\sum_{i=1}^n \epsilon_i\sigma(\mathbf{W}f(\mathbf{x}_i))\right\|\right)$$

$$\stackrel{(i)}{=} \mathbb{E}_{\mathbf{x},\epsilon} \sup_{f\in\mathcal{F}, \|\mathbf{w}\|\leq R} g\left(\left|\sum_{i=1}^n \epsilon_i\sigma(\mathbf{w}f(\mathbf{x}_i))\right|\right), \tag{65}$$

where (i) follows from the second equation in the proof of Lemma 1 in [8]. Noting that $g(x) \geq 0$, we have $g(|x|) \leq g(x) + g(-x)$, and hence (65) is upper-bounded by

$$\mathbb{E}_{\mathbf{x},\epsilon} \sup_{f\in\mathcal{F}, \|\mathbf{w}\|\leq R} g\left(\sum_{i=1}^n \epsilon_i\sigma(\mathbf{w}f(\mathbf{x}_i))\right) + \mathbb{E}_{\mathbf{x},\epsilon} \sup_{f\in\mathcal{F}, \|\mathbf{w}\|\leq R} g\left(-\sum_{i=1}^n \epsilon_i\sigma(\mathbf{w}f(\mathbf{x}_i))\right),$$

which, using the symmetry of the distribution of the Rademacher random variable $\epsilon_i$, is equal to

$$2\mathbb{E}_{\mathbf{x},\epsilon} \sup_{f\in\mathcal{F}, \|\mathbf{w}\|\leq R} g\left(\sum_{i=1}^n \epsilon_i\sigma(\mathbf{w}f(\mathbf{x}_i))\right). \tag{66}$$

Recall that $g$ is increasing and convex and note that $\sigma(0) = 0$. Then, based on the equation (4.20) [1] in [12], we further upper-bound (66) by

$$2\mathbb{E}_{\mathbf{x},\epsilon} \sup_{f\in\mathcal{F}, \|\mathbf{w}\|\leq R} g\left(LR\left\|\sum_{i=1}^n \epsilon_i f(\mathbf{x}_i)\right\|\right) = 2\mathbb{E}_{\mathbf{x},\epsilon} \sup_{f\in\mathcal{F}} \exp\left(\eta R^2 L^2 \left\|\sum_{i=1}^n \epsilon_i f(\mathbf{x}_i)\right\|^2\right). \tag{67}$$

The proof of the second result follows from that of Lemma 2 in [8]. $\qquad\square$

Next, we provide the main part of the proof.

**Case 1: parameter set $\mathcal{W}_F$.** Let $F_i(\mathbf{x}) = \sigma_{i-1}(\mathbf{W}_{i-1}\sigma_{i-2}(\cdots\sigma_1(\mathbf{W}_1\mathbf{x})))$. Then, for any $\lambda > 0$,

$$
n\mathcal{R}_n(\mathcal{F}_{nn},\mu) = \mathbb{E}_{\mathbf{x},\epsilon} \sup_{F_{d-1},\mathbf{w}_d,\mathbf{W}_{d-1}} \left| \sum_{i=1}^{n} \epsilon_i \mathbf{w}_d^T \sigma_{d-1}(\mathbf{W}_{d-1} F_{d-1}(\mathbf{x}_i)) \right|
$$

$$
= \sqrt{\frac{1}{\lambda}\log\left(\exp\lambda\left(\mathbb{E}_{\mathbf{x},\epsilon}\sup_{F_{d-1},\mathbf{w}_d,\mathbf{W}_{d-1}}\left|\sum_{i=1}^{n}\epsilon_i\mathbf{w}_d^T\sigma_{d-1}(\mathbf{W}_{d-1}F_{d-1}(\mathbf{x}_i))\right|\right)^2\right)}
$$

$$
\overset{(i)}{\le} \sqrt{\frac{1}{\lambda}\log\left(\mathbb{E}_{\mathbf{x},\epsilon}\exp\left(\lambda\sup_{F_{d-1},\mathbf{w}_d,\mathbf{W}_{d-1}}\left|\sum_{i=1}^{n}\epsilon_i\mathbf{w}_d^T\sigma_{d-1}(\mathbf{W}_{d-1}F_{d-1}(\mathbf{x}_i))\right|\right)^2\right)}
$$

$$
\overset{(ii)}{=} \sqrt{\frac{1}{\lambda}\log\left(\mathbb{E}_{\mathbf{x},\epsilon}\sup_{F_{d-1},\mathbf{w}_d,\mathbf{W}_{d-1}}\exp\left(\lambda\left|\sum_{i=1}^{n}\epsilon_i\mathbf{w}_d^T\sigma_{d-1}(\mathbf{W}_{d-1}F_{d-1}(\mathbf{x}_i))\right|^2\right)\right)}
$$

$$
\le \sqrt{\frac{1}{\lambda}\log\left(\mathbb{E}_{\mathbf{x},\epsilon}\sup_{F_{d-1},\mathbf{W}_{d-1}}\exp\left(\lambda M_F(d)^2\left\|\sum_{i=1}^{n}\epsilon_i\sigma_{d-1}(\mathbf{W}_{d-1}F_{d-1}(\mathbf{x}_i))\right\|^2\right)\right)}
$$

$$
\overset{(iii)}{\le} \sqrt{\frac{1}{\lambda}\log\left(\mathbb{E}_{\mathbf{x},\epsilon}\sup_{F_{d-1}}\exp\left(\lambda M_F(d)^2 M_F(d-1)^2 L_{d-1}^2\left\|\sum_{i=1}^{n}\epsilon_i F_{d-1}(\mathbf{x}_i)\right\|^2\right)\right)}, \quad (68)
$$

where (i) follows from Jensen's inequality, (ii) follows from the fact that the function $\exp(\lambda x^2)$ is strictly increasing over $[0,\infty]$ and (iii) follows from Lemma 4. Repeating the step (iii) in (68) for $d-1$ times yields

$$
n\mathcal{R}_n(\mathcal{F}_{nn},\mu) \le \sqrt{\frac{1}{\lambda}\log\left(2^{d-1}\mathbb{E}_{\mathbf{x},\epsilon}\left(e^{\lambda M^2\left\|\sum_{i=1}^{n}\epsilon_i\mathbf{x}_i\right\|^2}\right)\right)}. \quad (69)
$$

where we define $M := \prod_{i=1}^{d} M_F(i)\prod_{i=1}^{d-1} L_i$. Next, we upper-bound the following term from (69)

$$
\mathbb{E}_{\mathbf{x},\epsilon}\exp\left(\lambda M^2\left\|\sum_{i=1}^{n}\epsilon_i\mathbf{x}_i\right\|^2\right) = \mathbb{E}_\epsilon\left(\mathbb{E}_\mathbf{x}\exp\left(\lambda M^2\left\|\sum_{i=1}^{n}\epsilon_i\mathbf{x}_i\right\|^2\right)\bigg|\epsilon_1,...,\epsilon_n\right) \quad (70)
$$

Conditioned on $\epsilon_1,...,\epsilon_n$, we define $\mathbf{z} = \sum_{i=1}^{n}\epsilon_i\mathbf{x}_i$. Recall that each $\mathbf{x}_i$ is a sub-Gaussian random vector with variance $\tau^2$ and mean $\mathbf{u}$. Thus, we have, for any vector $\mathbf{a}\in\mathbb{R}^h$,

$$
\mathbb{E}_\mathbf{z} e^{\mathbf{a}^T(\mathbf{z}-\mathbb{E}\mathbf{z})} = \prod_{i=1}^{n}\mathbb{E}_{\mathbf{x}_i}e^{\mathbf{a}^T\epsilon_i(\mathbf{x}_i-\mathbb{E}\mathbf{x}_i)} \le e^{\|\mathbf{a}\|^2 n\tau^2/2},
$$

which implies that $\mathbf{z}$ is a sub-Gaussian random vector with variance $n\tau^2$ and mean $\mathbf{u}_z = \mathbf{u}\sum_{i=1}^{n}\epsilon_i$. Then, using Lemma 3 in the proof of Theorem 5, we obtain, for any $0\le\lambda M^2\le 1/(2n\tau^2)$,

$$
\mathbb{E}_\mathbf{z}\exp\left(\lambda M^2\|\mathbf{z}\|^2\right) \le \exp\left(n\tau^2 h\lambda M^2 + \frac{n^2\tau^4 h\lambda^2 M^4 + \|\mathbf{u}_z\|^2\lambda M^2}{1-2n\tau^2\lambda M^2}\right)
$$

$$
\overset{(i)}{\le} \exp\left(n\tau^2 h\lambda M^2 + \frac{n^2\tau^4 h\lambda^2 M^4}{1-2n\tau^2\lambda M^2}\right)\exp\left(\frac{\Gamma_{\text{uB}}^2\lambda M^2\left|\sum_{i=1}^{n}\epsilon_i\right|^2}{1-2n\tau^2\lambda M^2}\right),
$$

where (i) follows from the fact that $\|\mathbf{u}\|^2\le\Gamma_{\text{uB}}^2$. Pick $\lambda = (1-2n\tau^2\lambda M^2)/(4\Gamma_{\text{uB}}^2 n M^2)$. Then, we have $\lambda M^2\le 1/(2n\tau^2)$, and

$$
\mathbb{E}_\mathbf{z}\exp\left(\lambda M^2\|\mathbf{z}\|^2\right) \le \exp\left(n\tau^2 h\lambda M^2\left(1+\frac{\tau^2}{4\Gamma_{\text{uB}}^2}\right)\right)\exp\left(\frac{\left|\sum_{i=1}^{n}\epsilon_i\right|^2}{4n}\right),
$$

which, in conjunction with (70), yields

$$\mathbb{E}_{\mathbf{x},\epsilon} \exp\left(\lambda M^2 \left\|\sum_{i=1}^{n} \epsilon_i \mathbf{x}_i\right\|^2\right) \leq \exp\left(n\tau^2 h\lambda M^2 \left(1 + \frac{\tau^2}{4\Gamma_{\mathrm{uB}}^2}\right)\right) \mathbb{E}_\epsilon \exp\left(\frac{\left|\sum_{i=1}^{n} \epsilon_i\right|^2}{4n}\right). \quad (71)$$

Based on the equation (1) in [17], we have

$$\mathbb{P}\left(\left|\sum_{i=1}^{n} \epsilon_i\right| \geq \sqrt{n}\delta\right) \leq 2e^{-\delta^2/2},$$

which implies that

$$\mathbb{P}\left(\exp\left(\frac{\left|\sum_{i=1}^{n} \epsilon_i\right|^2}{4n}\right) \geq \exp\left(\frac{\delta^2}{4}\right)\right) = \mathbb{P}\left(\left|\sum_{i=1}^{n} \epsilon_i\right| \geq \sqrt{n}\delta\right) \leq 2e^{-\delta^2/2}. \quad (72)$$

Defining a random variable $\mathbf{Y} = \exp\left(\left|\sum_{i=1}^{n} \epsilon_i\right|^2/(4n)\right)$, and letting $t = \exp\left(\delta^2/4\right) \geq 1$, the inequality (72) can be rewritten as $\mathbb{P}(\mathbf{Y} \geq t) \leq 2/t^2$. Then, we can obtain

$$\mathbb{E}_\epsilon \exp\left(\frac{\left|\sum_{i=1}^{n} \epsilon_i\right|^2}{4n}\right) = \mathbb{E}(\mathbf{Y}) = \int_1^\infty \mathbb{P}(\mathbf{Y} \geq t)\,\mathrm{d}t \leq \int_1^\infty \frac{2}{t^2}\,\mathrm{d}t = 2,$$

which, combined with (71), implies that

$$\mathbb{E}_{\mathbf{x},\epsilon} \exp\left(\lambda M^2 \left\|\sum_{i=1}^{n} \epsilon_i \mathbf{x}_i\right\|^2\right) \leq 2\exp\left(n\tau^2 h\lambda M^2 \left(1 + \frac{\tau^2}{4\Gamma_{\mathrm{uB}}^2}\right)\right). \quad (73)$$

Combining (69) and (73) and recalling that $\lambda = 1/(4\Gamma_{\mathrm{uB}}^2 nM^2 + 2n\tau^2 M^2)$, we have

$$n\mathcal{R}_n(\mathcal{F}_{nn}, \mu) \leq \sqrt{n}\Gamma_{\mathrm{uB}} M\sqrt{6d\log 2 + 5h/4},$$

which further yields

$$\mathcal{R}_n(\mathcal{F}_{nn}, \mu) \leq \frac{\Gamma_{\mathrm{uB}} M\sqrt{6d\log 2 + 5h/4}}{\sqrt{n}}.$$

**Case 2: parameter set $\mathcal{W}_{1,\infty}$.** Using an approach similar to (68) and applying Lemma 4, we obtain, for any $\lambda > 0$,

$$n\mathcal{R}_n(\mathcal{F}_{nn}, \mu) \leq \frac{1}{\lambda} \log\left(2^{d-1}\mathbb{E}_{\mathbf{x},\epsilon} \exp\left(\lambda M \left\|\sum_{i=1}^{n} \epsilon_i \mathbf{x}_i\right\|_\infty\right)\right), \quad (74)$$

where $M = \prod_{i=1}^{d} M_{1,\infty}(i) \prod_{i=1}^{d-1} L_i$. Letting $x_{ij}$ be the $j^{th}$ coordinate of $\mathbf{x}_i$, the expectation term in (74) can be rewritten as

$$\mathbb{E}_{\mathbf{x},\epsilon} \exp\left(\lambda M \max_j \left|\sum_{i=1}^{n} \epsilon_i x_{ij}\right|\right) \leq \sum_{j=1}^{h} \mathbb{E}_{\mathbf{x},\epsilon} \exp\left(\lambda M \left|\sum_{i=1}^{n} \epsilon_i x_{ij}\right|\right)$$

$$\leq \sum_{j=1}^{h} \mathbb{E}_{\mathbf{x},\epsilon}\left(\exp\left(\lambda M \sum_{i=1}^{n} \epsilon_i x_{ij}\right) + \exp\left(-\lambda M \sum_{i=1}^{n} \epsilon_i x_{ij}\right)\right)$$

$$= 2\sum_{j=1}^{h} \mathbb{E}_{\mathbf{x},\epsilon} \exp\left(\lambda M \sum_{i=1}^{n} \epsilon_i x_{ij}\right) \stackrel{(i)}{=} 2\sum_{j=1}^{h} \mathbb{E}_\epsilon\left(\prod_{i=1}^{n} \mathbb{E}_{\mathbf{x}} \exp\left(\lambda M \epsilon_i x_{ij}\right)\Big|\epsilon_1, ..., \epsilon_n\right)$$

$$\stackrel{(ii)}{\leq} 2\sum_{j=1}^{h} \mathbb{E}_\epsilon\left(\prod_{i=1}^{n} \exp\left(M^2\lambda^2\tau^2/2\right)\right) = 2h\exp\left(\lambda^2 M^2 n\tau^2/2\right), \quad (75)$$

where (i) follows from the fact that $\mathbf{x}_1, ..., \mathbf{x}_n$ are independent and (ii) follows from the definition of the sub-Gaussian random variable. Combining (74) and (75) yields

$$n\mathcal{R}_n(\mathcal{F}_{nn}, \mu) \leq \frac{d\log 2 + \log h}{\lambda} + \lambda\frac{M^2 n\tau^2}{2} \stackrel{(i)}{=} M\tau\sqrt{2n}\sqrt{d\log 2 + \log h}$$

$$\stackrel{(ii)}{\leq} M\Gamma_{\mathrm{uB}}\sqrt{2n}\sqrt{d\log 2 + \log h},$$

where (i) is obtained by picking $\lambda = \sqrt{2(d\log 2 + \log h)/(M^2 n\tau^2)}$ and (ii) follows from the fact that $\tau \leq \Gamma_{\mathrm{uB}}$. Then, the proof is complete.

# D Proof of Theorem 4

The proof is similar to that of Theorem 6. First, we have

$$|d_{\mathcal{F}_{nn}}(\mu, \nu) - d_{\mathcal{F}_{nn}}(\hat{\mu}, \hat{\nu})| \leq F(\mathbf{x}_1, ..., \mathbf{x}_n, \mathbf{y}_1, ..., \mathbf{y}_m).$$

where the function $F$ is defined in (55). We start with the case when the parameter set is $\mathcal{W}_F$, and then adapt the proof to the parameter set $\mathcal{W}_{1,\infty}$.

**Case 1: parameter set $\mathcal{W}_F$.** Using an approach similar to (58) and (59), we can obtain,

$$|F(..., \mathbf{x}_i, ...) - F(..., \mathbf{x}_i', ...)| \leq \prod_{i=1}^{d} M_F(i) \prod_{i=1}^{d-1} L_i \|\mathbf{x}_i - \mathbf{x}_i'\|/n \leq 2\Gamma_{\mathrm{B}} \prod_{i=1}^{d} M_F(i) \prod_{i=1}^{d-1} L_i/n$$

$$|F(..., \mathbf{y}_i, ...) - F(..., \mathbf{y}_i', ...)| \leq \prod_{i=1}^{d} M_F(i) \prod_{i=1}^{d-1} L_i \|\mathbf{y}_i - \mathbf{y}_i'\|/m \leq 2\Gamma_{\mathrm{B}} \prod_{i=1}^{d} M_F(i) \prod_{i=1}^{d-1} L_i/m,$$

which, using the standard McDiarmid inequality [16], implies

$$\mathbb{P}\left(F(\mathbf{x}_1, ..., \mathbf{x}_n, ...., \mathbf{y}_m) - \mathbb{E}\, F(\mathbf{x}_1, ..., \mathbf{x}_n, ...., \mathbf{y}_m) \geq \epsilon\right) \leq \exp\left(\frac{-\epsilon^2 mn}{2K^2(m+n)}\right) \tag{76}$$

where $K := \Gamma_{\mathrm{B}} \prod_{i=1}^{d} M_F(i) \prod_{i=1}^{d-1} L_i$. In order to upper-bound the expectation term in (76), using an approach similar to (62) yields

$$\mathbb{E}\, F(\mathbf{x}_1, ..., \mathbf{x}_n, ...., \mathbf{y}_m) \leq 2\mathcal{R}_n(\mathcal{F}_{nn}, \mu) + 2\mathcal{R}_m(\mathcal{F}_{nn}, \nu). \tag{77}$$

Let $\delta = \exp\left(-\epsilon^2 mn/(2K^2(m+n))\right)$. Combining (76) and (77) implies that , with probability at least $1 - \delta$,

$$F(\mathbf{x}_1, ..., \mathbf{x}_n, ...., \mathbf{y}_m) \leq 2\mathcal{R}_n(\mathcal{F}_{nn}, \mu) + 2\mathcal{R}_m(\mathcal{F}_{nn}, \nu)$$

$$+ \Gamma_{\mathrm{B}} \prod_{i=1}^{d} M_F(i) \prod_{i=1}^{d-1} L_i \sqrt{2\log\frac{1}{\delta}} \sqrt{\frac{1}{n} + \frac{1}{m}}, \tag{78}$$

where the Rademacher complexity

$$\mathcal{R}_n(\mathcal{F}_{nn}, \mu) = \mathbb{E}_{\mathbf{x}, \epsilon} \sup_{f \in \mathcal{F}_{nn}} \left|\frac{1}{n}\sum_{i=1}^{n} \epsilon_i f(\mathbf{x}_i)\right| = \mathbb{E}_{\mathbf{x}}\left(\mathbb{E}_\epsilon\left(\sup_{f \in \mathcal{F}_{nn}} \left|\frac{1}{n}\sum_{i=1}^{n} \epsilon_i f(\mathbf{x}_i)\right|\right) \bigg| \mathbf{x}_1, ..., \mathbf{x}_n\right). \tag{79}$$

Conditioned on $\mathbf{x}_1, ..., \mathbf{x}_n$, we define

$$\hat{\mathcal{R}}_n(\mathcal{F}_{nn}) = \mathbb{E}_\epsilon \sup_{f \in \mathcal{F}_{nn}} \left|\frac{1}{n}\sum_{i=1}^{n} \epsilon_i f(\mathbf{x}_i)\right|.$$

Let $F_i(\mathbf{x}) = \sigma_{i-1}(\mathbf{W}_{i-1}\sigma_{i-2}(\cdots\sigma_1(\mathbf{W}_1\mathbf{x})))$. Then, we can obtain

$$n\hat{\mathcal{R}}_n(\mathcal{F}_{nn}) = \mathbb{E}_\epsilon \sup_{F_{d-1}, \mathbf{w}_d, \mathbf{W}_{d-1}} \left|\sum_{i=1}^{n} \epsilon_i \mathbf{w}_d^T \sigma_{d-1}(\mathbf{W}_{d-1} F_{d-1}(\mathbf{x}_i))\right|$$

$$= \frac{1}{\lambda}\log\left(\exp\lambda\left(\mathbb{E}_\epsilon \sup_{F_{d-1}, \mathbf{w}_d, \mathbf{W}_{d-1}} \left|\sum_{i=1}^{n} \epsilon_i \mathbf{w}_d^T \sigma_{d-1}(\mathbf{W}_{d-1} F_{d-1}(\mathbf{x}_i))\right|\right)\right)$$

$$\overset{(i)}{\leq} \frac{1}{\lambda}\log\left(\mathbb{E}_\epsilon \sup_{F_{d-1}, \mathbf{w}_d, \mathbf{W}_{d-1}} \exp\lambda\left(\left|\sum_{i=1}^{n} \epsilon_i \mathbf{w}_d^T \sigma_{d-1}(\mathbf{W}_{d-1} F_{d-1}(\mathbf{x}_i))\right|\right)\right)$$

$$\leq \frac{1}{\lambda}\log\left(\mathbb{E}_\epsilon \sup_{F_{d-1}, \mathbf{W}_{d-1}} \exp\lambda\left(M_F(d)\left\|\sum_{i=1}^{n} \epsilon_i \sigma_{d-1}(\mathbf{W}_{d-1} F_{d-1}(\mathbf{x}_i))\right\|\right)\right). \tag{80}$$

where (i) follows from the Jensen's inequality. Recall that $\|\mathbf{x}_i\| \leq \Gamma_{\mathrm{B}}$ for $i = 1, 2, ..., n$. Then, combining (80) and the steps of the proof of Theorem 1 in [8] yields

$$\hat{\mathcal{R}}_n(\mathcal{F}_{nn}) \leq \frac{\Gamma_{\mathrm{B}} \prod_{i=1}^{d} M_F(i) \prod_{i=1}^{d-1} L_i(\sqrt{2d\log 2} + 1)}{\sqrt{n}},$$

which, in conjunction with (78) and (79), yields the first result of Theorem 4.

**Case 2: parameter set** $\mathcal{W}_{1,\infty}$. Using an approach similar to (78), we can obtain

$$F(\mathbf{x}_1, ..., \mathbf{x}_n, ...., \mathbf{y}_m) \leq 2\mathcal{R}_n(\mathcal{F}_{nn}, \mu) + 2\mathcal{R}_m(\mathcal{F}_{nn}, \nu)$$
$$+ \Gamma_{\mathrm{B}} \prod_{i=1}^{d} M_{1,\infty}(i) \prod_{i=1}^{d-1} L_i \sqrt{2 \log \frac{1}{\delta}} \sqrt{\frac{1}{n} + \frac{1}{m}}, \qquad (81)$$

Then, similarly to the proof of Theorem 2 in [8], we can obtain

$$\hat{\mathcal{R}}_n(\mathcal{F}_{nn}) \leq \frac{2\Gamma_{\mathrm{B}} \prod_{i=1}^{d} M_{1,\infty}(i) \prod_{i=1}^{d-1} L_i \sqrt{d+1+\log h}}{\sqrt{n}},$$

which, in conjunction with (81), implies the second result of Theorem 4.

## Footnotes

[1] Although this result requires $\sigma(\cdot)$ to be 1-Lipchitz, it can be directly extended to any Lipchitz constant $L > 0$ by replacing the Lipchitz constant 1 with $L$ in its proof.