[Reviews · NeurIPS 2018]

Reviewer 1



This paper analyzes the problem of estimating the neural net distance which is the optimization objective in GAN. The authors obtained improved upper and lower bounds on the minimax risk in estimating the distance, and for ReLU networks they showed that they match not only in terms of the sample sizes (which is quite trivial) but also the parameters of the network (which is more interesting). The analysis of quite solid and involves refined analysis of the Rademacher complexity, and the careful application of the Le Cam method for minimax lower bound. I think this paper helps people understand better the instrinsic discriminating power of ReLU networks, which is a good contribution.

Reviewer 2



* summary This paper deals with the generalization bound of the neural net distance. Under the standard assumptions, the minimax lower bound of the generalization error was provided for two classes of probability distributions. In addition, tight upper bounds for generalization ability were established by using a new McDiarmid's inequality and a tighter Rademacher complexity of neural networks. * review: A new McDiarmid's inequality is thought to be the main theoretical contribution. It enables us to evaluate the generalization error under the probability distribution with an unbounded support. Also, the Rademacher complexity of neural networks under sub-Gaussian distribution is interesting. A minor defect is the lack of numerical experiments to confirm the theoretical findings. Overall, theoretical results of this paper are solid and interesting. comments: line 181: the pattern of the equation of C(P_B) is not clear. Do all M_F(*)s in the second term have the negative sign? line 191: "the distribution class P_{uB}" may be a typo. I believe P_{uB} should be replaced with P_B. line 273: the definition of the random variable, epsilon, should be clearly mentioned, though the Rademacher complexity is a common tool.

Reviewer 3



This paper considers the minimax estimation problem of neural net distance. The problem originates from the generative adversarial networks (GANs). In particular, the paper established the lower bound on the minimax estimation for neural net distance which seems to the first result of this kind. The authors then derived an upper bound on the estimation error which matches the minimax lower bound in terms of the order of sample size, and the norm of the parameter matrices. However, there is a gap of $\sqrt{d}$ or $\sqrt{d+h}$ which remains as an open question. The main techniques root from a critical lemma called Le Cam's methods in nonparametric statistics and the recent paper [26]. The upper bound used the Rademacher complexity and the extension of McDiarmid's inequality for unbounded support distributions and sub-gaussian. Overall, the results are novel and very interesting in the setting of neural net distance. Several comments below: 1. The form of neural net distance is very similar to the MMD work in [26] where the same techniques (Le Cam's method) were used for deriving the minimax lower bound. It would be nice to explain the difference between this study and the techniques used in [26]. 2. There is a gap between the minimax lower bound and the upper bound used. Any comments on this? Is this because the hypothesis space of deep network is too large for the neural net distance? 3. The notation $\mathal{F}_{nn}$ for representing the function class of neural network could be confusing since here $n$ denotes the number of samples I am quite satisfied with the authors' feedback. It is a good paper.